# Effectiveness and optimal dosage of exercise training for chronic non-specific neck pain: A systematic review with a narrative synthesis

Jonathan Price [1,2], Alison Rushton [2], Isaak Tyros [2,3], Vasileios Tyros [3], Nicola R Heneghan [2]*

**1** Musculoskeletal Physiotherapy Services, Birmingham Community Healthcare NHS Foundation Trust, Birmingham, England, United Kingdom, **2** Centre of Precision Rehabilitation for Spinal Pain (CPR Spine) School of Sport, Exercise and Rehabilitation Sciences, College of Life and Environmental Sciences, University of Birmingham, Birmingham, England, United Kingdom, **3** Edgbaston Physiotherapy Clinic, Birmingham, England, United Kingdom

* n.heneghan@bham.ac.uk

**Data Availability Statement:** All relevant data are within the manuscript and its Supporting Information files.

## Abstract

### Background

Clinical guidelines make vague recommendations as to exercise training (ET) type and dosage to manage chronic non-specific neck pain (CNSNP).

### Objective

To synthesise evidence on the effectiveness of different ET programmes to reduce CNSNP and associated disability, and whether dosage affects outcomes.

### Methods

A systematic review and data synthesis was conducted according to a published registered protocol (PROSPERO CRD42018096187). A sensitive topic-based search was conducted of CINAHL, MEDLINE, EMBASE, PEDro, grey literature sources and key journals from inception to 6th January 2020 for randomised controlled trials, investigating ET for CNSNP or disability. Two reviewers independently completed eligibility screening, data extraction, risk of bias assessment (Cochrane Risk of Bias Tool) and rated the overall strength of evidence using Grading of Recommendations Assessment, Development and Evaluation. Data was tabulated for narrative synthesis and grouped by intervention, outcome and time point to compare across studies.

### Results

Twenty-six trials from 3990 citations (n = 2288 participants) investigated fifteen ET programmes. High RoB and low sample sizes reduced evidence quality. Clinical heterogeneity prevented meta-analyses. A range of ET programmes reduce pain/disability in the short term (low to moderate evidence). Pillar exercises reduce pain/disability in the intermediate term (low level evidence). Moderate to very large pain reduction is found with ET packages

**Funding:** This work was completed as part of a Clinical Health Research MRes at the University of Birmingham, UK funded by Health Education England and National Institute for Health Research (HEE/ NIHR ICA Programme Pre-doctoral Clinical Academic Fellowship, Mr Jonathan Price, ICA-PCAF-2018-01-117). The views expressed in this publication are those of the author(s) and not necessarily those of the NHS, the NIHR or the Department of Health and Social Care. Isaak Tyros and Vasileios Tyros are employed by Edgbaston Physiotherapy Clinic. The funder provided support in the form of salaries for authors IT and VT, but did not have any additional role in the study design, data collection and analysis, decision to publish, or preparation of the manuscript.The specific roles of these authors are articulated in the 'author contributions' section.

**Competing interests:** Jonathan Price, Alison Rushton and Nicola R Heneghan declare that they have no competing interests. Isaak Tyros and Vasileios Tyros are employed by the commercial company Edgbaston Physiotherapy Clinic. This does not alter our adherence to PLOS ONE policies on sharing data and materials.

that include motor control + segmental exercises (low to moderate evidence). No high-quality trials investigated long term outcomes. Increased frequency of motor control exercises and progressively increased load of pillar exercise may improve effectiveness.

## Conclusions

Motor control + segmental exercises are the most effective ET to reduce short term pain/disability, but long-term outcomes have not been investigated. Optimal motor control + segmental exercise variables and dosage is unknown and requires clarification. An adequately powered, low RoB trial is needed to evaluate the effectiveness and optimal dosage of motor control + segmental on long term outcomes.

## Trial registration

PROSPERO CRD42018096187

# 1 introduction

## 1.1 Rationale

Neck pain is highly prevalent affecting up to 50% of the population annually and now ranked 4th for global disability [1–3]. Clinical guidelines recommend neck pain and disability is treated using a multimodal package of care with exercise as an integral component [4, 5]. Despite short term benefits of exercise, long term effectiveness is unclear as 70% of individuals will develop recurring or persistent chronic non-specific neck pain (CNSNP) [6–8]. To reduce the recurring symptoms experts agree that the paramount priority for neck pain research should be to better understand the effectiveness of exercise interventions and how different exercise variables, such as dosage can maximise effects [9].

Clinical guidelines and systematic reviews provide recommendations based on moderate evidence that "exercise" or "strengthening and endurance" exercise have small to large effects on pain and disability but provide little detail to the type of exercise to be used in clinical practice [4, 5, 10–12]. The studies cited within these guidelines and systematic reviews describe multiple different exercise training (ET) programmes aimed at improving neuromuscular function or motor capacity of the neck and shoulder musculature. The ET programmes reported consist of various exercises such as cervical isometrics [13], cervical concentric/eccentric training using pulley systems or weights [14], upper limb training using dumbbells [15] or deep neck flexor/extensor rehabilitation [16, 17] all resulting in different changes in spinal function e.g. craniocervical flexion performance, cervical flexion strength [18, 19]. Within practice the intended effect of exercise on spinal function should inform the design of ET programmes [20, 21]. No systematic review has yet investigated the effectiveness of different ET programmes based on the intended effect on spinal function reducing CNSNP or disability.

Moreover exercise dosage, a component of exercise prescription is poorly described in neck pain clinical guidelines is exercise dosage [22]. Manipulating dosage (duration, frequency, intensity) has significant effects on physical outcomes such as strength, power, hypertrophy in a sports and performance setting [23, 24]. Evidence suggests higher dosages improving patient reported outcomes in neck pain of varying durations [25, 26] however neck pain clinical guidelines do not provide dosage recommendations and no evidence synthesis has been undertaken. Therefore the optimal dosage to improve pain and disability in a chronic neck

pain population is not known and further investigation is recommended by Cochrane [10], experts [9, 21, 27, 28] and professional bodies [29]. Initially, a systematic review is required to synthesise the current evidence to investigate the effectiveness of different dosages of ET programmes in reducing CNSNP or disability to guide future research.

## 1.2 Objectives

The primary objective of this systematic review is to synthesise the current evidence to investigate the effectiveness of ET programmes categorised by their intended effect on spinal function in reducing CNSNP and disability. A secondary objective is to investigate whether ET dosage affects outcomes. There are two hypotheses to this systematic review:

1. Exercise training programmes categorised by their intended effect on spinal function have different effects on chronic non-specific neck pain and disability

2. Exercise training programmes of different dosages have different effects on chronic non-specific neck pain and disability

# 2 methods

## 2.1 Protocol and registration

This systematic review was conducted according to a pre-defined, registered (PROSPERO CRD42018096187) and published protocol, and reported using Preferred Reporting Items for Systematic Reviews and Meta-Analyses (PRISMA)(S1 Appendix) [30–32]. PROSPERO was amended 27/2/2019 to include another reviewer. The term resistance training has been changed from the protocol to "exercise training" as suggested by our patient and public involvement group to reflect exercise where the goal is to improve neuromuscular function or motor capacity of the neck and shoulder musculature. This has not changed the inclusion or exclusion criteria for included trials.

## 2.2 Eligibility criteria

Eligibility criteria were developed by scoping searches and PICOS.

 **2.2.1 Participants.** Aged 18–70 years experiencing ≥3 months non-specific neck pain [33]. Specific pathologies (whiplash associated disorder, headaches, cervical radiculopathy etc) were excluded [30].

 **2.2.2 Intervention.** Interventions considered ET and included in this synthesis were exercises targeted at the neck or shoulders where an individual applies a force against resistance (gravity, their own hands, an external object) to improve neuromuscular function or motor capacity. Motor control exercises were included providing resistance was applied using a biofeedback unit or gravity. Exercises requiring a therapists assistance or exercises for sensorimotor control disturbances (e.g. cervical joint position sense, oculomotor, gaze stability etc exercises) were excluded [34]. Stretching or aerobic training were excluded unless part of a warmup or cool down. Combined ET and another intervention (e.g. manual therapy, education etc) programmes were included if possible, to derive data specifically for the ET component.

 **2.2.3 Comparator.** Any comparator was included e.g. other exercise, other therapies, or no treatment.

 **2.2.4 Outcome measures.** Any patient reported measure of neck pain [e.g. Visual Analogue Scale (VAS)) and/or neck functional disability (e.g. Neck Disability Index, (NDI)].

**2.2.5 Study design.** Randomised controlled clinical trials. Pilot or feasibility studies were excluded.

**2.2.6 Report eligibility.** Trials not written in English and protocols for trials not yet completed were excluded at full text and reported within the PRISMA flow diagram. There were no publication date restrictions.

## 2.3 Information sources

Electronic database searches were performed from inception to 06[th] January 2020 using CINAHL; EMBASE; MEDLINE; PubMed; PEDro; Index to Chiropractic Literature and TRIP. Unpublished literature was searched using Zetoc and OpenGrey [35]. Conference proceedings and articles in press/published ahead of print were searched in key journals Spine, European Spine Journal, Journal of Orthopaedic and Sports Physical Therapy, Strength and Conditioning Journal and The Journal of Strength and Conditioning. Key publishers (Elsevier, Springer, Wiley) were searched for articles published but not yet indexed in medical databases. Reference lists of all included citations were reviewed.

## 2.4 Search

JP (subject expertise) completed all searches. S2 Appendix contains database search strategies, keywords, and MeSH. Citations were stored and de-duplicated in Endnote X9 [36].

## 2.5 Study selection

Two independent reviewers (JP/IT) (subject expertise) performed title/abstract and full text screening with disagreements resolved through discussion. A third reviewer (NH) (subject and methodological expertise) was available if necessary.

## 2.6 Data collection process

Two independent reviewers (JP/VT) extracted data using Cochrane's data extraction form which was adapted and piloted on 5 randomly selected trials (S3 Appendix) [37]. Disagreements were resolved by discussion with a third reviewer (NH) available if necessary. Trial authors were contacted where data was missing or ambiguous.

## 2.7 Data items

Study and participant characteristics, outcome measures and results were extracted (see protocol for further details) [30, 33]. Follow up periods were recorded as immediate ($\leq$24 hours), short term ($>$24 hours $\leq$3 months), intermediate term ($>$3 months $<$ 12 months) and long term ($\geq$12 months) [10]. Intervention data was recorded using the TIDieR Checklist [38]. Grouping of ET programmes was based on the intended effect on spinal function using an expert derived exercise classification system [20].

## 2.8 Risk of bias

Two independent reviewers (JP/VT) assessed Risk of Bias (RoB) using the Cochrane RoB Tool [31, 35]. Piloting agreed interpretation of each domain as "unclear", "low" or "high" RoB (S4 Appendix). A third reviewer (NH) mediated disagreements and Cohen's *k* assessed inter-rater agreement [35]. Results were tabulated to evaluate RoB across studies and used to inform Grading of Recommendations, Assessment, Development and Evaluation ratings. Further exploration of RoB across studies (selective outcome reporting, publication bias) were planned where a minimum of 10 studies were included for meta-analysis [39, 40].

### 2.9 Summary measures

Continuous outcomes were analysed using Review Manager 5.3 to calculate mean difference or standardised mean difference (with 95% CIs) where measurement scales varied [35]. Effect sizes were reported using Cohen's *d* where ≤0.2 trivial; >0.2 small; >0.5 moderate; >0.8 large; >1.3 very large [41]. Clinically importance differences were established *a priori* as a mean difference of >1/10 VAS for pain and >5/50 NDI for disability [10, 42].

### 2.10 Synthesis of results

Overall quality of evidence for each ET programme reducing pain/disability for all follow up periods was rated using Grading of Recommendations, Assessment, Development and Evaluation (GRADE) [35, 43, 44]. Quality of evidence was assessed as 'high' 'moderate' 'low' or 'very low' by two independent reviewers (JP/VT) following piloting (S5 Appendix).

Post treatment means and standard deviations were extracted owing to multiple trials not reporting change from baseline standard deviations [35]. To ensure lower scores reflect a "better" outcome for all scales, mean scores for any outcome measure using a reverse scale (i.e. where a lower score reflects a "worse" outcome) were multiplied by -1 [35]. The furthest time point from randomisation was extracted where multiple follow ups within 1-time period existed. Imputation methods were used to estimate mean and standard deviation values from minimum and maximum values, first and third quartiles, medians and sample size where authors could not provide clarification [45].

As per the protocol a meta-analysis was planned providing low clinical and statistical heterogeneity existed assessed using the Table of Characteristics and $I^2 < 50\%$ and $p > 0.10$ respectively [30, 35]. Where meta-analysis was not possible, a narrative synthesis provided summaries of the evidence [30, 46, 47]. Trials were grouped by ET classification and the effect on pain/disability was described narratively with forest plots reporting standardised mean differences plus 95% CI without a pooled estimate. The impact of dosage on effectiveness was investigated between clinically homogenous trials using a narrative description and a potential moderator variable table including standardised mean differences plus 95% CI [46]. Dosage analysis was completed for classifications of ET where the overall quality of evidence was rated moderate or high.

### 2.11 Additional analyses

Sensitivity analysis was planned by repeating meta-analysis excluding high RoB trials and those with missing data.

### 2.12 Patient involvement

The study was conceived from the views of patients with spinal complaints from our clinical working. Patients suggested the term exercise training to reflect any exercise where the goal is to improve neuromuscular function. Findings will be disseminated to patients via patient workshops.

## 3 Results

### 3.1 Study selection

From 3990 citations, 275 full texts were screened from which 33 citations met eligibility criteria (Fig 1) (S6 Appendix for excluded studies). Agreement between reviewers was 100% at each stage. Multiple reports of the same trial were collated for Waling et al., 2002 [48–50], Bobos

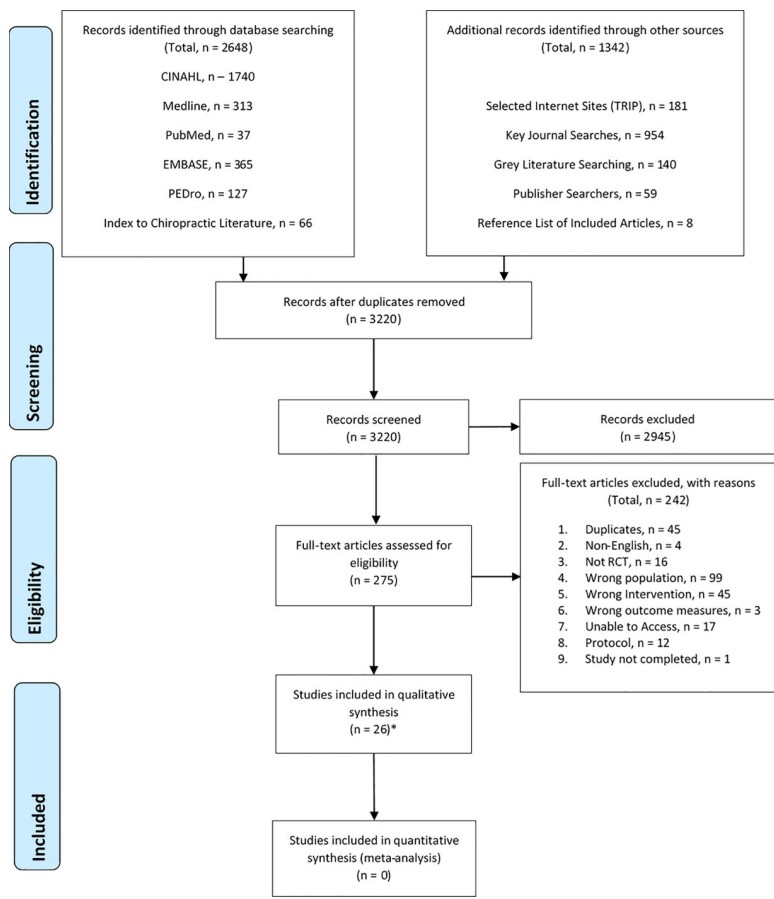

**Fig 1. PRISMA flow diagram of study selection process.** *33 citations. Abbreviations: *RCT*—Randomised Controlled Trial.

et al., 2016 [51, 52], Ylinen et al., 2007 [53–56], and Chiu et al., 2005 [57, 58]. Data extraction was completed for 26 trials [13, 14, 49, 51, 54, 57, 59–78].

**Table 1. Exercise training classifications, definitions, and example exercises.**

| Exercise Training Classification | Definition | Exercise reported in Trials |
|---|---|---|
| Motor Control | Exercises intended to retrain co-ordination of cervical musculature or sequential segmental control of spinal movement using submaximal effort | Craniocervical flexion in supine with biofeedback unit; Craniocervical extension/flexion/rotation in 4-point kneeling |
| Pillar | Exercises intended to develop the ability of the spine to maintain a neutral position | Cervical isometric flexion/extension/rotation/lateral flexion using hand as resistance/pulley system/resistance bands; Cervical isometric flexion against gravity in sitting |
| Segmental | Exercises intended to develop the ability of the spine to endure the production, transference, or absorption of forces through the performance of sequential segmental movements | Cervical flexion/extension/lateral flexion using pulley system; Cervical flexion in supine; Cervical flexion from a position of extension in sitting; Cervical extension in prone/4-point kneeling; Cervical retraction against resistance band |
| Upper Limb | Exercises intended to change the neuromuscular performance of the shoulder or shoulder girdle musculature | Resisted row; Triceps press, Shoulder press; Lat pull down; Shrugs; Bicep curls; Fly's; Pull overs; Chest press; Scapular retraction with resistance band; Horizontal pull a part; Serratus anterior punches; Glenohumeral abduction with dumbbells |

**Table 2. Brief intervention details used in each trial.**

| Author., Date | Brief Intervention Information *Treatment Category Intervention Description* |
|---|---|
| Chung et al., 2018 [13] | **1.** *Motor Control Exercises* |
| | ***Intervention Description:*** *(1) Warm up–neck stretches (2) Craniocervical flexion in supine (3) Cool down–neck stretches* **Dosage:** *Duration– 8 weeks; Frequency–three per week; Intensity–(2) 10x10 second holds, 3–5 seconds rest* |
| | **2.** *Pillar Exercises* |
| | ***Intervention Description:*** *Week 1 - (1) Warm up–neck stretches (2) Cervical isometrics in supine\* (3) Cool down–neck stretches; Weeks 2–8 –(1) (4) Cervical isometric flexion/ extension/rotation/lateral flexion in sitting using hand as resistance (3)* **Dosage:** *Duration– 8 weeks; Frequency–three per week; Intensity–(2)(4) 10-15x10 second holds, 15 seconds rest* |
| Jordan et al., 1998 [14] | **1.** *Segmental + Upper Limb Resistance Training Exercises + Another Intervention* |
| | ***Intervention Description:*** *(1) Single 1.5-hour neck school (2) Cervical flexion using neck exercise unit (3) Cervical extension/lateral flexion (4) Shoulder, scapular, chest exercises using hand-held weights\*, lat pull down (5) Cool down–Static Bike; HEP (6) 5 strengthening exercises for neck and shoulders\* (7) 3 stretching exercises* **Dosage:** *Duration– 6 weeks; Frequency–Supervised sessions—twice per week, HEP–not reported; Intensity–Supervised sessions–(2) 1x12 @ 30% MVC (3) 3x12 @ 30% MVC (4)(6) not reported* |
| | **2.** *Passive Physiotherapy + Another Intervention* |
| | ***Intervention Description:*** *(1) Single 1.5-hour neck school (2) Passive physiotherapy; HEP (3) 5 strengthening exercises for neck and shoulders\* (4) 3 stretching exercises* **Dosage:** *Duration– 6 weeks; Frequency–Supervised sessions—twice per week, HEP–not reported; Intensity–(3)(4) not reported* |
| | **3.** *Manipulation + Another Intervention* |
| | ***Intervention Description:*** *(1) Single 1.5-hour neck school (2) Manipulation; HEP (3) 5 strengthening exercises for neck and shoulders\* (4) 3 stretching exercises* **Dosage:** *Duration– 6 weeks; Frequency–Supervised sessions—twice per week, HEP–not reported; Intensity–(3) (4) not reported* |
| Waling et al., 2002 [49] | **1.** *Upper Limb Resistance Training Exercises* |
| | ***Intervention Description:*** *(1) Warm up–no details (2) Concentric phase only of Row, triceps press, shoulder press, lat pull down* **Dosage:** *Duration– 10 weeks; Frequency–three per week; Intensity–Weeks 1–4 (2) 2x12RM; Weeks 5–10 (2) 3x12RM* |
| | **2.** *Aerobic Exercise + Upper Limb Resistance Training Exercise* |
| | ***Intervention Description:*** *(1) Warm up–no details (2) Arm ergometer (3) Upper limb exercises using rubber expanders* **Dosage:** *Duration– 10 weeks; Frequency–three per week; Intensity–(2) 110–120 BPM (3) 3-4x30-35RM* |
| | **3.** *Body Awareness Training* |
| | ***Intervention Description:*** *(1) Warm up–no details (2) Body awareness training* **Dosage:** *Duration– 10 weeks; Frequency–three per week; Intensity–NA* |
| | **4.** *Education and Stress Reduction* |
| | ***Intervention Description:*** *(1) Stress management education* **Dosage:** *Duration– 10 weeks; Frequency–once per week; Intensity–NA* |

(*Continued*)

**Table 2.** (Continued)

| Author., Date | Brief Intervention Information *Treatment Category Intervention Description* |
|---|---|
| Bobos et al., 2016 [51] | **1.** *Motor Control + Segmental Exercises + Another Intervention* |
| | *Intervention Description:* (1) *Craniocervical Flexion in Supine with feedback unit; (2) Cervical flexion/extension in supine/prone (Craniocervical neutral not maintained throughout movement); (3) Nodding from prone position; (4) Nodding in standing with head against wall; (5) Standard exercise leaflet with stretches and isometric exercises* ***Dosage:*** *Duration– 7 weeks; Frequency Supervised sessions–twice per week, HEP–not reported; Intensity–not reported* |
| | **2.** *General Neck Exercise + Another Intervention* |
| | *Intervention Description:* (1) *Posterior head movement from sitting position; (2) Posterior head movement from supine position; (3) Movement in all directions in prone; (4) Cat-camel; (5) Standard exercise leaflet with stretches and isometric exercises* ***Dosage:*** *Duration– 7 weeks; Frequency Supervised sessions–twice per week, HEP–not reported; Intensity–not reported* |
| | **3.** *Another Intervention* |
| | *Intervention Description:* (1) *Standard exercise leaflet with stretches and isometric exercises* ***Dosage:*** *Duration– 7 weeks; Frequency HEP–not reported; Intensity–not reported* |
| Ylinen et al., 2007 [54] | **1.** *Pillar + Upper Limb Resistance Training Exercises + Another Intervention* |
| | *Intervention Description:* (1) *Cervical isometric flexion/extension/oblique flexion using theraband (maintain cervical neutral while moving trunk against resistance of theraband) (2) Dumbbell shrugs, shoulder press, curls, bent over rows, flys, pull overs (3) Lower body exercise (4) Stretching (5) Aerobic exercise (6) Manual therapy (7) 12-day neck school* ***Dosage:*** *Duration–Supervised sessions– 2 weeks, HEP– 12 months; Frequency–Supervised sessions–six per week, HEP–three per week; Intensity–(1) 1x15 @ 80% maximal isometric strength (2) 1x15RM* |
| | **2.** *Segmental + Upper Limb Resistance Training Exercises + Another Intervention* |
| | *Intervention Description:* (1) *Cervical flexion in supine (Craniocervical neutral not maintained throughout movement) (2) Dumbbell shrugs, shoulder press, curls, bent over rows, flys, pull overs (3) Lower body exercise (4) Stretching (5) Aerobic exercise (6) Manual therapy (7) 12-day neck school* ***Dosage:*** *Duration–Supervised sessions– 2 weeks, HEP– 12 months; Frequency–Supervised sessions–six per week, HEP–three per week; Intensity–(1) 3x20 @ weight of head (2) 3x20 @ 2kg* |
| | **3.** *Stretching*§ |
| | *Intervention Description:* (1) *Stretching* ***Dosage:*** *Duration–Supervised sessions– 3 days, HEP– 12 months; Frequency–HEP–three per week; Intensity–NA* |
| Chiu et al., 2005 [57] | **1.** *Motor Control + Segmental Exercises + Another Intervention* |
| | *Intervention Description:* (1) *20 mins infrared and neck care advice; (2) Craniocervical flexion in supine (3) Cervical flexion and extension warm up (4) Cervical flexion and extension using multi cervical rehabilitation unit* ***Dosage:*** *Duration– 6 weeks; Frequency–twice per week; Intensity–(2) 22–30 mmHg, 10 second holds, 15 seconds rest, 10 mins (3) 1x15 @ 20% peak isometric strength (4) 3x8–12 @ 30% peak isometric strength, 5 mins rest between sets* |
| | **2.** *TENs + Another Intervention* |
| | *Intervention Description:* (1) *20 mins infrared and neck care advice; (2) 30 mins TENs* ***Dosage:*** *Duration– 6 weeks; Frequency–twice per week* |
| | **3.** *Another Intervention* |
| | *Intervention Description:* (1) *20 mins infrared and neck care advice* ***Dosage:*** *Duration– 6 weeks; Frequency–twice per week* |

*(Continued)*

**Table 2.** (Continued)

| Author., Date | Brief Intervention Information *Treatment Category Intervention Description* |
| --- | --- |
| Javanshir et al., 2015 [59] | **1.** *Motor Control Exercises* |
| | *Intervention Description:* (1) Craniocervical flexion in supine **Dosage:** Duration– 10 weeks; Frequency–Supervised sessions–three per week, HEP–Three per day; Intensity–(1) 10x10 second holds, 10 seconds rest |
| | **2.** *Segmental Exercises* |
| | *Intervention Description:* (1) Cervical flexion in supine with craniocervical neutral maintained **Dosage:** Duration– 10 weeks; Frequency–Supervised sessions–three per week, HEP–Three per day; Intensity–(1) Stage 1 (Weeks 0–2) 1x12-15RM, Stage 2 (Weeks 3–10) 3x15 @ original 12RM, 1 min rest between sets |
| O'Leary et al., 2007 [60] | **1.** *Motor Control Exercises* |
| | *Intervention Description:* (1) Craniocervical flexion in supine **Dosage:** Duration–single session; Frequency–NA; Intensity–(1) 10x10 second holds, 10 seconds rest |
| | **2.** *Segmental Exercises* |
| | *Intervention Description:* (1) Cervical flexion in supine with craniocervical neutral maintained **Dosage:** Duration–single session; Frequency–NA; Intensity–(1) 3x10 @ 12RM, 3 second holds, 2 seconds rest between reps, 30 seconds rest between sets |
| Rudolfsson et al., 2014 [61] | **1.** *Co-ordination Exercises* |
| | *Intervention Description:* (1) Neck Co-ordination exercises **Dosage:** Duration– 11 weeks; Frequency–twice per week; Intensity–NA |
| | **2.** *Pillar + Upper Limb Resistance Training Exercises* |
| | *Intervention Description:* (1) Cervical isometric flexion/lateral flexion/rotation using pulley system (maintain cervical neutral while moving trunk against resistance of pulley system) (2) Shoulder press, chest press and seated row **Dosage:** Duration– 11 weeks; Frequency–twice per week; Intensity–Sessions 1–3 (1) 1x15, 3 second holds, 11–13 BORG Scale (2) 2x15, 11–13 BORG Scale; Session 4 – 1RM testing; Session 5–11 (1) 1x15 @ 60% 1RM measured at session 4, RPE 13 (2) 2X12 @ 60% 1RM measured at session 4; Session 11 – 1RM testing; Session 12 onwards (1) 1x8 @ 75% 1RM measured at session 11 (2) 2x8 @ 75% 1RM measured at session 11 |
| | **3.** *Massage* |
| | *Intervention Description:* (1) Massage **Dosage:** Duration– 11 weeks; Frequency–twice per week; Intensity–NA |
| Yildiz et al., 2017 [62] | **1.** *Upper Limb Resistance Training + Segmental + Motor Control Exercises + Another Intervention* |
| | *Intervention Description:* (1) Push up plus in an oblique position, scapular retraction with theraband, lateral pull down with theraband (2) Cervical flexion in supine with craniocervical neutral maintained (3) Cervical retraction against resistance (4) Craniocervical flexion in supine (5) Cervical stretches (6) Manual therapy **Dosage:** Duration– 6 weeks; Frequency–Supervised sessions—once per week, HEP–twice per day; Intensity–(1) 3x10 (2) 2x10 (3) 2x10 @ 6/10 BORG Scale (4) 10x10 second holds |
| | 2. Control Group—(15)—*Segmental + Motor Control Exercises + Another Intervention* |
| | *Intervention Description:* (1) Cervical flexion in supine with craniocervical neutral maintained (2) Cervical retraction against resistance (3) Craniocervical flexion in supine (4) Cervical stretches (5) Manual therapy **Dosage:** Duration– 6 weeks; Frequency–Supervised sessions—once per week, HEP–twice per day; Intensity–(1) 2x10 (2) 2x10 @ 6/10 BORG Scale (3) 10x10 second holds |

*(Continued)*

**Table 2.** (Continued)

| Author., Date | Brief Intervention Information *Treatment Category Intervention Description* |
|---|---|
| Borisut et al., 2013 [63] | **1.** *Segmental Exercises* |
| | *Intervention Description:* (1) Cervical Flexion/Extension in Supine/Prone (Craniocervical neutral not maintained throughout movement) **Dosage:** Duration– 12 weeks; Frequency– once per day; Intensity–Stage 1 (Weeks 1–4) 1x12 – 15RM, 1 min rest between sets; Stage 2 (Weeks 5–12) 3x15 @ original 12RM, 1 min rest between sets |
| | **2.** *Motor Control Exercises* |
| | *Intervention Description:* (1) Craniocervical flexion in supine **Dosage:** Duration– 12 weeks; Frequency–once per day; Intensity– 22–30 mmHg, 15x10 second holds, 10 seconds rest |
| | **3.** *Motor Control + Segmental Exercises* |
| | *Intervention Description:* (1) Cervical flexion/extension in supine/prone (Craniocervical neutral not maintained throughout movement); (2) Craniocervical flexion in supine **Dosage:** Duration– 12 weeks; Frequency–once per day; Intensity–(1)—Stage 1 (Weeks 1–4) 1x12 – 15RM, 1 min rest between sets; Stage 2 (Weeks 5–12) 3x15 @ original 12RM, 1 min rest between sets; (2) 22–30 mmHg, 15x10 second holds, 10 seconds rest |
| | **4.** *No Treatment* |
| Falla et al., 2013 [64] | **1.** *Motor Control + Segmental Exercises* |
| | *Intervention Description:* Stage 1 (1) Craniocervical flexion in supine; (2) Craniocervical extension/flexion/rotation in prone propped on elbows; Stage 2 (3) Cervical flexion with craniocervical flexion in supine; (4) Cervical extension with craniocervical neutral in 4-point kneeling **Dosage:** Duration– 8 weeks; Frequency–Supervised sessions—once per week, HEP– Twice per day; Intensity–Stage 1 (1–2)–Unclear, Stage 2 (3–4) 1x15, 3 second holds |
| | **2.** *No Treatment* |
| Li et al., 2017 [65] | **1.** *Pillar Exercises* |
| | *Intervention Description:* (1) Standard instruction booklet about office ergonomics (2) Warm Up–General cervical and shoulder active range of movement exercises (3) Cervical isometric flexion/extension/lateral flexion using theraband attached to fixed stable object (Trunk movement was not performed to create resistance against the band) **Dosage:** Duration– 6 weeks; Frequency–three per week; Intensity–(3) Weeks 1–2–8–12 @ 30% maximal strength, 5 seconds rest; Weeks 3–4–8–12 @ 50% maximal strength measured at week 2, 5 seconds rest; Weeks 5–6–8–12 @ 70% maximal strength measured at week 4, 5 seconds rest |
| | **2.** *Pillar Exercises* |
| | *Intervention Description:* (1) Standard instruction booklet about office ergonomics (2) Warm Up–General cervical and shoulder active range of movement exercises (3) Cervical isometric flexion/extension/lateral flexion using theraband attached to fixed stable object (Trunk movement was not performed to create resistance against the band) **Dosage:** Duration– 6 weeks; Frequency–three per week; Intensity–(3) 8–12 @ 70% maximal strength, 5 seconds rest |
| | **3.** *Education* |
| | *Intervention Description:* (1) Standard instruction booklet about office ergonomics (2) Weekly discussions about workplace ergonomics, stress management, relaxation meditation and diet **Dosage:** Duration– 6 weeks; Frequency–once per week; Intensity–NA |
| Viljanen et al., 2003 [66] | **1.** *'XX' + Upper Limb Resistance Training Exercises* |
| | *Intervention Description:* (1) Neck and shoulder exercises using dumbbells* **Dosage:** Duration– 12 weeks; Frequency–three per week; Intensity–(1) not reported |
| | **2.** *Relaxation Techniques* |
| | *Intervention Description:* (1) Relaxation training using progressive relaxation methods, autogenic training, functional relaxation and systematic desensitisation |
| | *Dosage:* Duration– 12 weeks; Frequency–three per week; Intensity–NA |
| | **3.** *No Treatment* |

(*Continued*)

**Table 2.** (Continued)

| Author., Date | Brief Intervention Information *Treatment Category Intervention Description* |
|---|---|
| Suvrannato et al., 2019 [67] | **1.** *Pillar Exercises (Therapist Assisted)* |
| | *Intervention Description:* (1) Isometric cervical extension isolating semispinalis cervicis with therapist assistance **Dosage:** Duration– 6 weeks; Frequency–Supervised sessions–twice per week; Intensity–(1) 3x10, 10 second holds, 30 seconds rest between sets |
| | **2.** *Motor Control Exercises* |
| | *Intervention Description:* (1) Craniocervical flexion in supine **Dosage:** Duration– 6 weeks; Frequency–Supervised sessions–twice per week, HEP–twice daily; Intensity–(1) 3x10, 10 second holds, 30 seconds rest between sets |
| | **3.** *Usual Care* |
| | *Intervention Description:* Any treatment deemed appropriate by physiotherapist including stretching, upper limb strengthening, manual therapy, electrotherapy. Intervention excluded craniocervical flexion in supine and isometric cervical extension **Dosage:** Duration– 6 weeks; Frequency–Supervised sessions– 10–12 over duration; Intensity–not reported |
| Shiravi et al., 2019 [68] | **1.** *Upper Limb Resistance Training Exercises* |
| | *Intervention Description:* (1) Overhead press (2) Horizontal pull aparts (3) Chest Press (4) Serratus anterior punches (5) Retraction + external rotation (6) Scapular protraction (7) XY (8) TYW **Dosage:** Duration– 6 weeks; Frequency–not reported; Intensity–not reported |
| | **2.** *No Treatment* |
| | **3.** *Upper Limb Resistance Training Exercises + Another Intervention*§ |
| | *Intervention Description:* (1) Overhead press (2) Horizontal pull aparts (3) Chest Press (4) Serratus anterior punches (5) Retraction + external rotation (6) Scapular protraction (7) XY (8) TYW (9) Abdominal controlled feedback with inferior glides, isometric low row, dynamic knee push ups, wall press and wall slides **Dosage:** Duration– 6 weeks; Frequency–not reported; Intensity–not reported |
| Gupta et al., 2010 [69] | **1.** *Motor Control + Segmental + Pillar Exercises* |
| | *Intervention Description:* Stage 1 (1) Craniocervical flexion in supine; Stage 2 (2) Cervical extension and return to neutral while maintaining craniocervical flexion; Stage 3 (3) Cervical extension and return to neutral with isometric holds throughout range, while maintaining craniocervical flexion **Dosage:** Duration– 6 weeks; Frequency–Once per day; Intensity–(1) (3) 10x10 second holds at target level; (2)—Unclear |
| | **2.** *Pillar Exercises* |
| | *Intervention Description:* (1) Cervical flexion/extension/lateral flexion isometrics–not clear how isometrics were performed **Dosage:** Duration– 6 weeks; Frequency–Once per day; Intensity– 30 mins |
| Gupta et al., 2013 [70] | **1.** *Motor Control Exercises* |
| | *Intervention Description:* (1) Craniocervical flexion in supine **Dosage:** Duration– 4 weeks; Frequency—not reported; Intensity–not reported |
| | **2.** *Pillar Exercises* |
| | *Intervention Description:* (1) Cervical isometrics in sitting using hand for resistance* **Dosage:** Duration– 4 weeks; Frequency—not reported; Intensity–not reported |
| Hingarajia et al., 2012 [71] | **1.** *Motor Control + Pillar Exercises* |
| | *Intervention Description:* (1) Craniocervical flexion in supine (2) Cervical isometrics flexion/extension/lateral flexion using hand as resistance **Dosage:** Duration– 4 weeks; Frequency–Supervised sessions—twice per week, HEP–Twice per day; Intensity–(1) 3x10, 10 second holds, 1 min rest between sets (2) 1x15, 10 second holds |
| | **2.** *Pillar Exercises* |
| | *Intervention Description:* (1) Cervical isometrics flexion/extension/lateral flexion using hand as resistance **Dosage:** Duration– 4 weeks; Frequency–Supervised sessions—twice per week, HEP–Twice per day; Intensity–(1) 1x15, 10 second holds |

*(Continued)*

**Table 2.** (Continued)

| Author., Date | Brief Intervention Information *Treatment Category Intervention Description* |
|---|---|
| Izquierdo et al., 2016 [72] | **1.** *Motor Control Exercises* |
| | *Intervention Description:* (1) Craniocervical flexion in supine **Dosage:** *Duration– 2 months; Frequency–Supervised sessions–once per week first 3 weeks then once every 2 weeks for 6 weeks, HEP–Twice per day; Intensity–(1) 10x10 second holds, 3–5 seconds rest* |
| | **2.** *Proprioception Exercises* |
| | *Intervention Description:* (1) Cervical proprioception exercises including head relocation, eye-follow, gaze stability, eye-head-co-ordination **Dosage:** *Duration– 2 months; Frequency–Supervised sessions–once per week first 3 weeks then once every 2 weeks for 6 weeks, HEP–Twice per day; Intensity–(1) 30 mins* |
| Kaur et al., 2018 [73] | **1.** *Motor Control Exercises* |
| | *Intervention Description:* (1) Craniocervical flexion in supine **Dosage:** *Duration–single session; Frequency–NA; Intensity–(1) 1 rep per 2 seconds for 1 min* |
| | **2.** *Manual Therapy* |
| | *Intervention Description:* (1) Passive grade III craniocervical flexion mobilisations **Dosage:** *Duration–single session; Frequency–NA; Intensity–(1) 1 rep per 2 seconds for 1 min* |
| Kim et al., 2016 [74] | **1.** *Motor Control Exercises* |
| | *Intervention Description:* (1) Craniocervical flexion in supine **Dosage:** *Duration– 4 weeks; Frequency–three per week; Intensity–(1) 10x10-15 second holds, 3–5 seconds rest* |
| | **2.** *Pillar Exercises + Another Intervention* |
| | *Intervention Description:* (1) Cervical isometric flexion/extension/lateral flexion (unclear how isometrics performed) (2) Cervical isometric flexion/extension/lateral flexion pushing head against ball that's on the wall (3) Cervical stretching **Dosage:** *Duration– 4 weeks; Frequency–three per week; Intensity–(1)(2) 10x10 second holds (3) 3x3-5, 10 second holds* |
| Kwan-Woo et al., 2016 [75] | **1.** *Thoracic Manipulation + Motor Control Exercises*§ |
| | *Intervention Description:* (1) Thoracic manipulation (2) Craniocervical flexion in supine **Dosage:** *Duration– 10 weeks; Frequency–three per week; Intensity–(2) 10x10 second holds, 5 seconds rest* |
| | **2.** *Motor Control Exercises* |
| | *Intervention Description:* (1) Craniocervical flexion in supine (2) Cool down–cervical stretches **Dosage:** *Duration– 10 weeks; Frequency–three per week; Intensity–(1) 10x10 second holds, 5 seconds rest* |
| | **3.** *General Active Range of Movement Exercises* |
| | *Intervention Description:* (1) General cervical active range of movement exercises **Dosage:** *Duration– 10 weeks; Frequency–three per week; Intensity–(1) 35 mins* |
| Randlov et al., 1998 [76] | **1.** *Segmental + Upper Limb Resistance Training Exercises* |
| | *Intervention Description:* (1) Warm Up–hot pack, static bike and stretching (2) Cervical flexion/extension in supine/prone (Craniocervical neutral not maintained throughout movement) (3) Arm abduction, scapular retraction and shoulder extension, supine shoulder flexion, wall push ups **Dosage:** *Duration– 12 weeks; Frequency–three per week; Intensity–(2)(3) 1x20* |
| | **2.** *Segmental + Upper Limb Resistance Training Exercises* |
| | *Intervention Description:* (1) Warm Up–hot pack, static bike and stretching (2) Cervical flexion/extension in supine/prone (Craniocervical neutral not maintained throughout movement) (3) Arm abduction, scapular retraction and shoulder extension, supine shoulder flexion, wall push ups **Dosage:** *Duration– 12 weeks; Frequency–three per week; Intensity–(2)(3) 5x20* |

(*Continued*)

**Table 2.** (Continued)

| Author., Date | Brief Intervention Information *Treatment Category Intervention Description* |
|---|---|
| Khan et al., 2014 [77] | **1.** *Pillar Exercises + Another Intervention* |
| | *Intervention Description*: *(1) Cervical isometric flexion/extension/lateral flexion/rotation using theraband (Unclear how theraband was used); HEP (2) General cervical range of movement* **Dosage**: *Duration– 12 weeks; Frequency–Supervised sessions–three per week, HEP–twice per day, 5 times per week; Intensity–(1)(2) 1x20* |
| | **2.** *Another Intervention* |
| | *Intervention Description*: *(1) General cervical range of movement; HEP (1)* **Dosage**: *Duration– 12 weeks; Frequency–Supervised sessions–three per week, HEP–twice per day, 5 times per week; Intensity–(1) 1x20* |
| Ulug et al., 2018 [78] | **1.** *Pilates + Another Intervention* |
| | *Intervention Description*: *(1) Hot pack, ultrasound, TEN's (2) Pilates* **Dosage**: *Duration Supervised sessions– 3 weeks, HEP– 6 weeks; Frequency–Supervised sessions—five per week, HEP–once per day; Intensity–(2) 2x10* |
| | **2.** *Yoga + Another Intervention* |
| | *Intervention Description*: *(1) Hot pack, ultrasound, TEN's (2) Yoga* **Dosage**: *Duration Supervised sessions– 3 weeks, HEP– 6 weeks; Frequency–Supervised sessions—five per week, HEP–once per day; Intensity–(2) 2x10* |
| | **3.** *Pillar Exercises + Another Intervention* |
| | *Intervention Description*: *(1) Hot pack, ultrasound, TEN's (2) Cervical isometric flexion/ lateral flexion in sitting using hand as resistance* **Dosage**: *Duration Supervised sessions– 3 weeks, HEP– 6 weeks; Frequency–Supervised sessions—five per week, HEP–once per day; Intensity–(2) 2x30, 5 second holds* |

Brief intervention description and dosage data for each trial. Full details can be found in S7 Appendix. Numbers in brackets cross reference intervention description with dosage information.

*No other details provided by authors,

§ Comparator does not meet inclusion criteria therefore treatment arm excluded from synthesis, XX—Exercises poorly described limiting classification. Abbreviations: HEP–Home Exercise Programme, MTrP–Myofascial Trigger Points, RM–Repetition Maximum, mmHg–Millimeter of Mercury, TENs–Transcutaneous Electrical Nerve Stimulation, CCF–Craniocervical Flexion, MVC–Maximal Voluntary Contraction, RPE–Rated of Perceived Exertion, BPM–Beats per Minute

## 3.2 Study characteristics

Table of characteristics and detailed intervention data are in S7 Appendix.

**3.2.1 Methods.** Trials were published between 1998–2019 (14 countries). Data clarification was required for 24/26 trials with nine authors responding [49, 51, 54, 59–63, 67]. Chiu et al reported conflicting results therefore data was extracted from the full trial report [57].

**3.2.2 Population.** A total of 2288 participants were included. Eligibility in trials varied, including symptom duration 3–12 months, minimum or maximum measures of pain/disability [51, 60–63, 65, 67–75], craniocervical flexion test performance ≤24mmHg [13, 69, 72–74] or scapular dyskinesis [62].

**3.2.3 Interventions.** Poor intervention reporting limited ET grouping based on the pre-defined classification which was subsequently adapted to motor control, pillar, segmental and upper limb exercises (Table 1). A total of 15 different ET programmes were identified: motor control exercises [13, 59, 60, 63, 67, 70, 72–75]; pillar exercises [13, 65, 69–71]; segmental exercises [59, 60, 63]; upper limb exercises [49, 68]; motor control + pillar exercises [71]; motor control + segmental exercises [63, 64]; motor control + segmental exercises + another intervention [51, 57]; motor control + segmental + pillar exercises [69]; pillar exercises + another intervention [77, 78]; pillar + upper limb exercises [61]; pillar + upper limb exercises + another

intervention; segmental + upper limb exercises [76]; segmental + upper limb exercises + another intervention [14, 54]; upper limb exercises + segmental + motor control exercises + another intervention [62] and upper limb exercise + another subgroup of neck exercises that were too poorly described to classify and referred to as "XX + Upper Limb"[66]. Table 2 provides a brief description of exercise training programmes and dosage.

ET was delivered via supervised sessions [13, 49, 54, 57, 60, 65, 68, 70, 73, 75, 76], a home exercise programme [63, 69, 74] or a combination of the two [14, 51, 59, 61, 62, 64, 66, 67, 71, 72, 77, 78], using a range of equipment including resistance bands [51, 54, 62, 65, 74, 77], biofeedback pressure units [13, 51, 57, 59, 60, 63, 64, 69, 70, 72–75], dumbbells [14, 54, 61, 66], weighted sandbags [59], balls [74] and resistance machines [14, 49, 61]. ET dosage was heterogenous across trials (duration: single session– 12 months; frequency: once per week–three times daily). Supervised treatment session time including non-ET interventions, ranged from 45 to 270 mins per week. Poor intervention reporting limited analysis of time spent completing ET at home or during supervised sessions. Intensity (sets, reps, rest, load) was poorly reported and varied according to ET.

**3.2.4 Comparators.** Comparator heterogeneity existed across trials (Table 2 and S7 Appendix) and prevented subgrouping.

**3.2.5 Outcome measures.** Pain and disability was used as a primary outcome measure in 8/26 trials [51, 54, 57, 62, 64–67] and measured with 15 and 8 different measurements tools respectively. Data was reported for immediate (3 trials) [60, 72, 73], short (23 trials) [13, 14, 49, 51, 54, 57, 59, 62–72, 74–78], intermediate (9 trials) [14, 49, 54, 57, 61, 65–67, 76], and long term effects (5 trials) [14, 49, 54, 66, 76].

## 3.3 Risk of bias

Complete agreement was achieved between reviewers. Only 3 trials had no high RoB domains (Table 3) [13, 59, 60]. Blinding of participants and reporting bias was high/unclear in 100% and 92% of studies respectively (Fig 2). Other sources of bias were baseline imbalances [51, 52, 64, 78], poor treatment fidelity [63, 69, 74], low compliance [66] and poor methodology reporting [68, 70, 71, 77].

## 3.4 Results of individual studies and synthesis of results

A narrative synthesis was performed as clinical heterogeneity (outcome measures, comparators) prevented meta-analysis. Comparator heterogeneity prevented the synthesis and analysis of evidence quality by contrasting each ET programme to subgroups of comparator interventions (e.g. Motor Control vs No Treatment). Therefore, the effectiveness of each ET programme has been narratively described against all reported comparators. Evidence quality for each ET programme is in S5 Appendix. Outcome measure heterogeneity limited summary measures to standardised mean differences plus 95% CI's. Means, standard deviations and standardised mean differences plus 95% CI's can be found in S8 Appendix.

## 3.5 Effectiveness of different exercise training programmes

**3.5.1 Motor control.** Motor control (MC) exercises reducing pain immediately was investigated in 3 trials with inconsistent findings (Fig 3). One trial (high RoB) found a large effect compared to manual therapy (SMD -1.09; 95%CI -1.91 to -0.36) [73] but two trials (1 high RoB, 1 low RoB) demonstrated no effect compared to other exercise [60, 72]. Based on very low-level evidence (GRADE) MC exercises are not effective reducing immediate pain.

**Table 3. Summary of risk of bias assessment.**

| Study | 1 | 2 | 3 | 4 | 5 | 6a | 6b | 7 | Summary risk of bias | Comment–High Risk Components |
|---|---|---|---|---|---|---|---|---|---|---|
| Chung et al., 2018 [13] | L | L | U | L | U | U | NA | U | Low (3) High (0) Unclear (4) | NA |
| Jordan et al., 1998 [14] | L | L | H | L | H | U | U | U | Low (3) High (2) Unclear (3) | High Components: 3, 5 (3: Participants would be aware of allocation due to significant differences in intervention, 5: No ITT) |
| Waling et al., 2002 [49] | H | H | H | U | H | U | U | L | Low (1) High (4) Unclear (3) | High Components: 1, 2, 3, 5 (1: Sequence determined by participants availability, 2: Investigators knew dates and times of interventions by which randomization would occur, 3: Participants would be aware of allocation due to significant differences in intervention, 5: >20% dropout at short term) |
| Bobos et al., 2016 [51] | L | L | H | H | H | L | NA | H | Low (3) High (4) Unclear (0) | High components: 3, 4, 5, 7 (3: Participants would be aware of allocation due to significant difference in intervention i.e supervised sessions vs HEP only; 4: Assessor not blinded; 5: No ITT analysis and reasons for dropouts not reported; 7: Baseline imbalance in NDI) |
| Ylinen et al., 2007 [54] | U | U | H | H | L | U | U | L | Low (2) High (2) Unclear (4) | High components: 3, 4 (3: Participants would be aware of allocation due to significant differences in intervention, 4: Assessor not blinded) |
| Chiu et al., 2005 [57] | L | L | H | L | L | U | NA | L | Low (5) High (1) Unclear (1) | High components: 3 (3: Participants would be aware of allocation due to significant differences in intervention) |
| Javanshir et al., 2015 [59] | L | L | U | L | L | U | NA | U | Low (4) High (0) Unclear (3) | NA |
| O'Leary et al., 2007 [60] | L | L | U | L | L | U | NA | L | Low (5) High (0) Unclear (2) | NA |
| Rudolfsson et al., 2014 [61] | L | L | H | L | H | U* | U* | L | Low (4) High (2) Unclear (2) | High components: 3, 5 (3: Participants would be aware of allocation due to significant differences in intervention, 5: >20% dropout rate) |
| Yildiz et al., 2017 [62] | L | L | U | L | H | L | NA | U | Low (4) High (1) Unclear (2) | High components: 5 (5: No ITT) |
| Borisut et al., 2013 [63] | L | U | H | L | L | U | NA | H | Low (3) High (2) Unclear (2) | High components: 3, 7 (3: Participants would be aware of allocation due to significant difference between intervention i.e no treatment vs exercise; 7: Treatment fidelity not assessed) |
| Falla et al., 2013 [64] | L | L | H | L | U | U | NA | H | Low (3) High (2) Unclear (2) | High components: 3, 7 (3: Participants would be aware of allocation due to significant differences in intervention i.e no treatment vs exercise; 7: Baseline Imbalance–SF– 36 statistically significantly lower in Intervention group) |
| Li et al., 2017 [65] | L | L | H | L | L | U | NA | L | Low (5) High (1) Unclear (1) | High components: 3 (3: Authors reported participants aware of allocation) |
| Viljanen et al., 2003 [66] | U | L | H | L | U | U | U | H | Low (2) High (2) Unclear (4) | High components: 3, 7 (3: Participants would be aware of allocation due to significant differences in intervention, 7: Poor compliance to intervention) |
| Suvrannato et al., 2019 [67] | L | L | H | L | L | U | NA | U | Low (4) High (1) Unclear (2) | High components: 3 (3: Participants would be aware of allocation due to significant differences in intervention) |
| Shiravi et al., 2019 [68] | U | L | H | U | H | U | NA | H | Low (1) High (3) Unclear (3) | High components: 3, 5, 7 (3: Participants would be aware of allocation due to significant differences in intervention, 5: Reasons for dropouts not reported and no ITT, 7: Consistent poor reporting of methods) |
| Gupta et al., 2010 [69] | U | U | H | L | L | U | NA | H | Low (2) High (2) Unclear (3) | High components: 3, 7 (3: Participants were aware of allocation; 7: Treatment fidelity not assessed) |
| Gupta et al., 2013 [70] | U | U | U | U | L | U | NA | H | Low (1) High (1) Unclear (5) | High components: 7 (7: Consistent poor reporting of methods) |
| Hingarajia et al., 2012 [71] | L | U | U | U | L | U | NA | H | Low (2) High (1) Unclear (4) | High components: 7 (7: Consistent poor reporting of methods) |
| Izquierdo et al., 2016 [72] | L | L | U | L | L | H | NA | U | Low (4) High (1) Unclear (2) | High components: 6a, (6a: Analysis of outcome measure completed but not reported in protocol) |
| Kaur et al., 2018 [73] | U | U | H | H | U | U | NA | H | Low (0) High (3) Unclear (4) | High components: 3, 4, 7 (3: Participants would be aware of allocation due to significant differences in intervention, 4: Assessor not blinded, 7: Baseline data such as demographics and group allocation not reported) |
| Kim et al., 2016 [74] | L | U | U | U | H | U | NA | H | Low (1) High (2) Unclear (4) | High components: 5, 7 (5: reason for dropouts not reported, no ITT, 7: Treatment fidelity not assessed) |

*(Continued)*

**Table 3.** (Continued)

| Study | 1 | 2 | 3 | 4 | 5 | 6a | 6b | 7 | Summary risk of bias | Comment–High Risk Components |
|---|---|---|---|---|---|---|---|---|---|---|
| Kwan-Woo et al., 2016 [75] | U | U | H | L | H | U | NA | U | Low (1) High (2) Unclear (4) | High components: 3, 5 (3: Participants would be aware of allocation due to significant differences in intervention, 5: Reason for dropouts not reported, no ITT, only baseline data for those completing treatment reported) |
| Randlov et al., 1998 [76] | L | U | U | U | H | U | U | U | Low (1) High (1) Unclear (6) | High Components: 5 (5: Imbalance in dropouts and no ITT) |
| Khan et al., 2014 [77] | L | U | U | U | U | U | NA | H | Low (1) High (1) Unclear (5) | High components: 7 (7: Consistent poor reporting of methods) |
| Ulug et al., 2018 [78] | L | U | U | U | H | U | NA | H | Low (1) High (2) Unclear (4) | High components: 5, 7 (1: 5: Reasons for dropouts not reported and no ITT, 7: Baseline Imbalance–Isometric group statistically significantly younger than other groups) |

*Author reports a further paper including all outcomes is in process and results are not yet available. Abbreviations: 1: Selection Bias (Random sequence generation); 2: Selection Bias (Allocation Concealment); 3: Performance Bias (Blinding of Participants); 4: Detection Bias (Blinding of Assessors); 5: Attrition Bias (Incomplete Outcome Data); 6a: Reporting Bias–Short Term Follow Up (Selective reporting, Identification of Primary Outcome Measures/Primary End Points); 6b: Reporting Bias–Long Term Follow Up (Selective reporting, Identification of Primary outcome measures/primary End Points); 7: Other (e.g. Fraud, Funding, Compliance, Treatment Fidelity, Baseline Imbalances)

MC exercises reducing immediate disability was investigated in one trial (high RoB) showing no effect compared to proprioceptive training [72]. Based on moderate level evidence (GRADE) MC exercises are not effective reducing immediate disability.

Short term pain and disability reduction was investigated in 8 trials with inconsistent findings (Fig 4 and Fig 5). Five trials (4 high RoB, 1 low RoB) found a moderate to very large effect on pain and disability compared to no treatment, usual care, general active range of movement (AROM) or pillar exercise [13, 63, 67, 70, 75]. Four trials (3 high RoB, 1 low RoB) found no effect on pain and disability compared to other ET and proprioceptive training [59, 63, 72, 74].

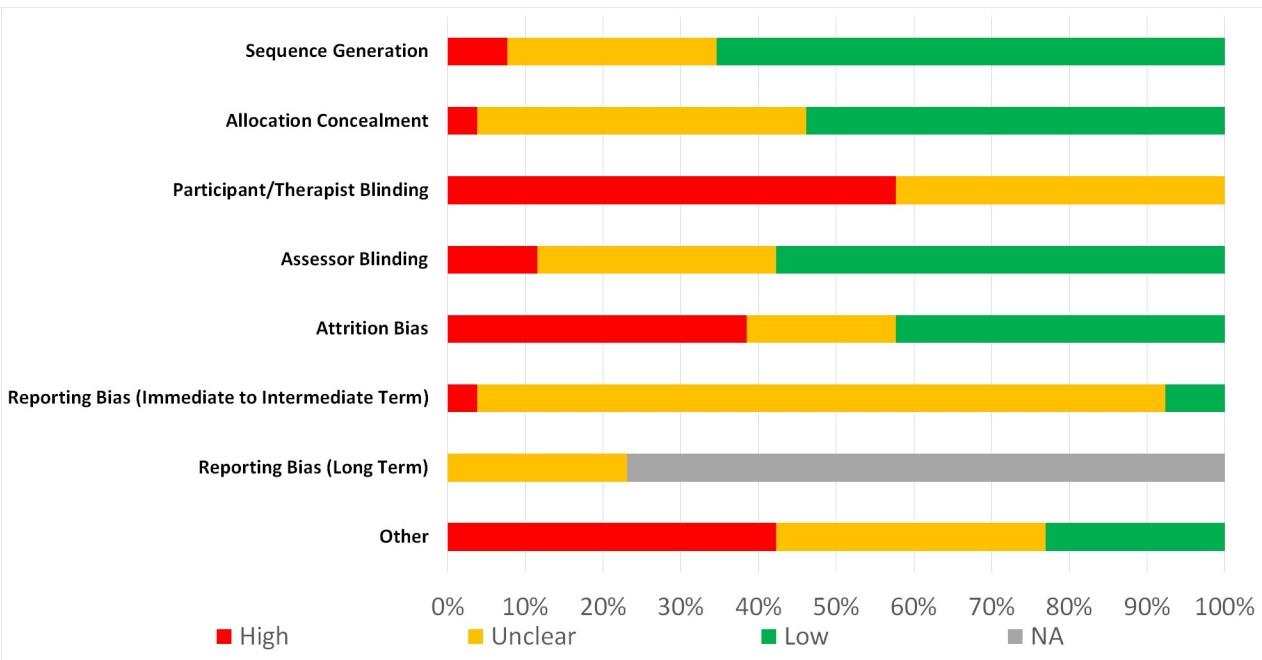

**Fig 2. Risk of bias graph by domain.** Review authors judgements about each risk of bias domain presented as percentages across all included studies. Grey areas represent clinical trials that did not assess long term outcomes preventing assessment of reporting bias at this time point.

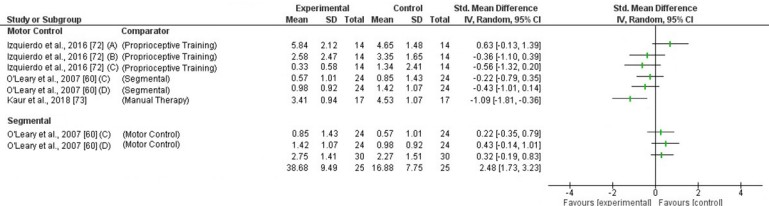

**Fig 3. Forest plot without a pooled estimate demonstrating the effectiveness of exercise training reducing pain at immediate term follow up.** Data is grouped by the exercise training programme considered to be the experimental intervention with comparator interventions identified for each trial. Where trials report multiple treatment arms, each comparator is reported separately. Exercise training programmes compared to other eligible exercise training programmes are presented twice so that both treatment arms can be considered the experimental intervention. All values reported using Numeric Rating Scale/Visual Analogue Scale unless otherwise indicated *A*–Maximum Visual Analogue Scale (VAS); *B*–Minimum *VAS; C*–VAS at Rest; *D*–VAS on activity. Abbreviations: *SD*—Standard Deviation; *CI*—Confidence Interval.

One trial (high RoB) found MC exercise to be less effective reducing pain than combinations of MC and segmental exercise [63]. Another trial (high RoB) found MC exercise to be less effective than pillar exercises performed with therapist's resistance for pain and disability reductions [67]. Based on moderate level evidence (GRADE) MC exercises are not effective reducing short term pain or disability.

MC exercises reducing intermediate pain and disability was investigated in one trial (high RoB) with inconsistent findings (Fig 6 and Fig 7). Effectiveness was demonstrated compared to usual care for disability reduction (SMD -4.72; 95%CI -6.04 to 3.39) but not for pain (SMD -0.47; 95%CI -1.15 to 0.18). MC exercise was found to be less effect than pillar exercises performed with therapist's resistance for pain and disability reduction (SMD 2.47; 95%CI 1.58 to 3.36 & SMD 3.53; 95%CI 2.44 to 4.60) [67]. Based on low level evidence (GRADE) MC exercises are not effective reducing intermediate term pain or disability.

**3.5.2 Pillar.** Pillar exercises reducing short term pain and disability was investigated in 5 trials with inconsistent findings (Fig 4 and Fig 5). One trial (high RoB) found very large improvements in pain and disability compared to education regardless of exercise dosage [65]. Four trials (3 high RoB, 1 low RoB) found pillar exercises to be less effective than other ET [13, 69–71]. Based on moderate level evidence (GRADE) pillar exercises are not effective reducing short term pain or disability.

Intermediate term pain and disability reduction was investigated in one trial (high RoB) (Fig 6 and Fig 7) [65]. The trial found a very large effect reducing pain (Fixed dosage: SMD -2.80; 95%CI -3.46 to -2.13; Progressive dosage: SMD -3.40; 95%CI -4.13 to -2.68) and disability (Fixed Dosage: SMD -2.10; 95%CI -2.68 to -1.51; Progressive Dosage: SMD -2.25; 95%CI -2.84 to -1.67) compared to education. Based on low level evidence (GRADE) pillar exercises are effective reducing intermediate term pain and disability.

**3.5.3 Segmental.** Segmental exercises reducing immediate pain was investigated in 1 trial (low RoB) showing no effect compared to MC exercise (Fig 3) [60]. Based on moderate level evidence (GRADE) segmental exercises are not effective reducing immediate pain.

Short term pain and disability reduction was investigated in 2 trials with inconsistent findings (Fig 4 and Fig 5). One trial (high RoB) found a very large effect reducing pain (SMD -2.14; 95%CI -2.84 to -1.43) and disability (SMD -3.90; 95%CI -4.86 to -2.93) compared to no treatment [63]. Two trials (1 high RoB, 1 low RoB) found no effect on pain or disability compared to other exercise [59, 63]. One trial (high RoB) found segmental exercise to be less effective reducing pain than combinations of MC and segmental exercise (SMD 2.48; 95%CI 1.73

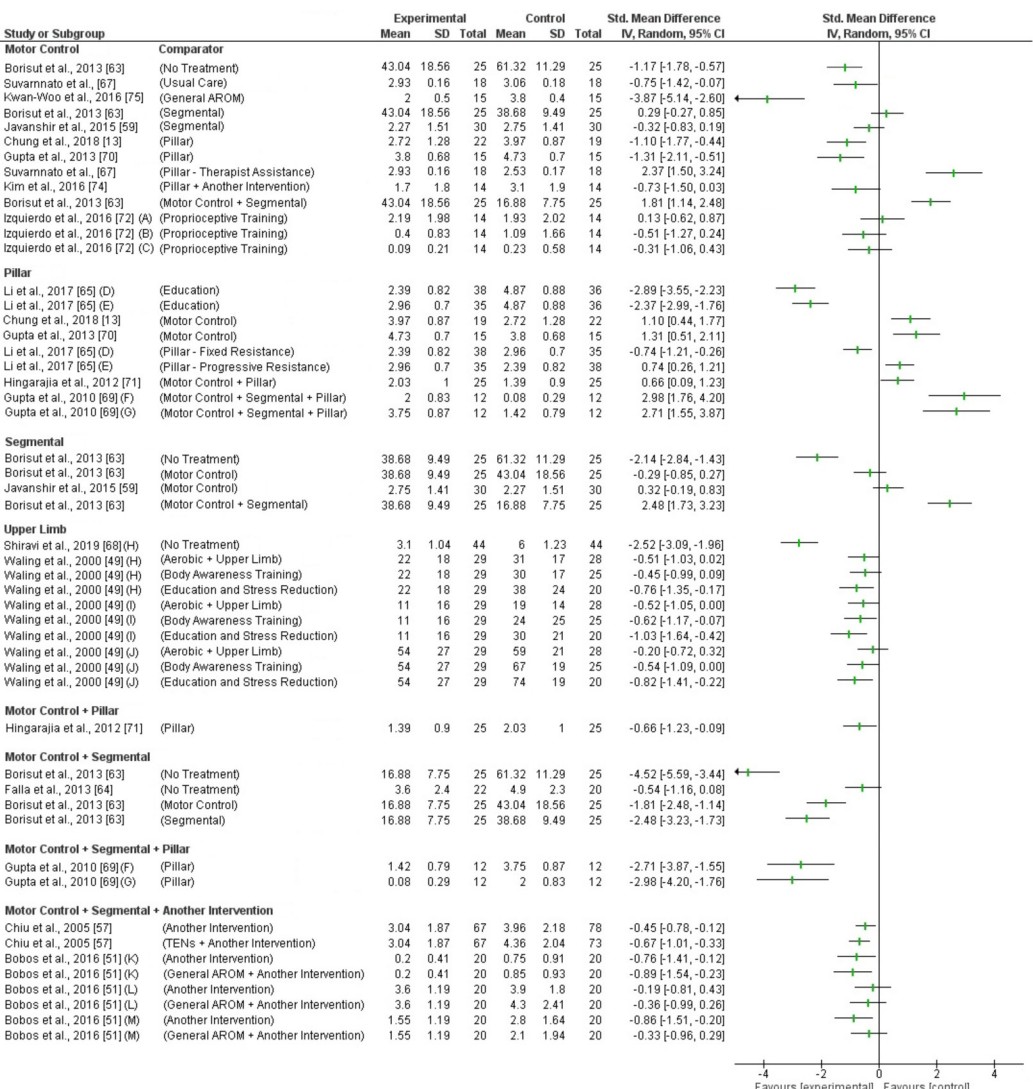

**Fig 4. Forest plot without a pooled estimate demonstrating the effectiveness of exercise training reducing pain at short term follow up.** Data is grouped by the exercise training programme considered to be the experimental intervention with comparator interventions identified for each trial. Where trials report multiple treatment arms, each comparator is reported separately. Exercise training programmes compared to other eligible exercise training programmes are presented twice so that both treatment arms can be considered the experimental intervention. All values reported using Numeric Rating Scale/Visual Analogue Scale unless otherwise indicated *A*–Maximum Visual Analogue Scale (VAS); *B*–Minimum *VAS; C*–VAS at Rest; *D*–Progressive load in experimental arm; *E*–Fixed load in experimental arm; *F*–VAS at rest; *G*–VAS on activity; *H*–VAS in general; *I*–VAS at present; *J*–VAS at worst; *K*–Numeric Rating Scale (NRS) at best; *L*–NRS at worst; *M*–NRS now. Abbreviations: *SD*—Standard Deviation; *CI*—Confidence Interval; *AROM*—Active Range of Movement.

to 3.23) [63]. Based on low level evidence (GRADE) segmental exercises are not effective reducing short term pain or disability.

**3.5.4 Upper limb.** Upper limb (UL) exercises reducing short, intermediate and long-term pain was investigated in 2 trials (high RoB) with inconsistent findings [49, 68]. In the short term (Fig 4) one trial found a very large effect when compared to no treatment (SMD -2.52; 95%CI -3.09 to -1.96) [68] and another trial found a moderate to large effect compared to education and stress reduction (General Pain: SMD; -0.76; 95%CI -1.35 to -0.17; Present Pain: SMD -1.03; 95%CI -1.64 to -0.42; Worst Pain: SMD -0.82; 95%CI -1.41 to -0.22), other exercise

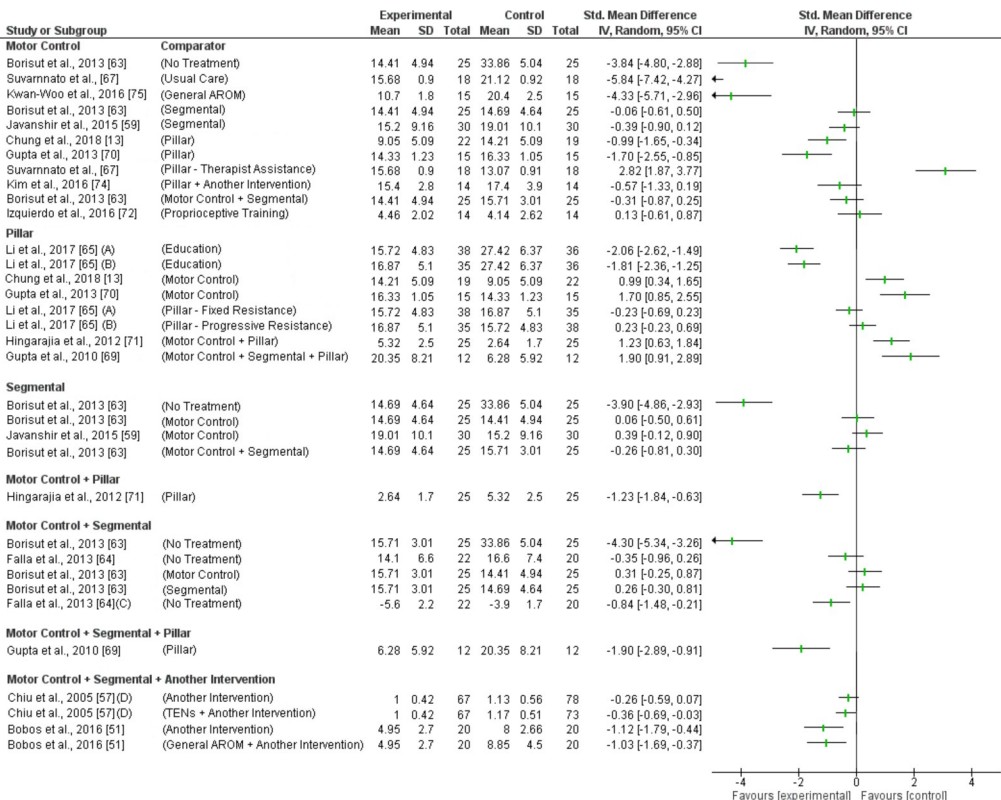

**Fig 5. Forest plot without a pooled estimate demonstrating the effectiveness of exercise training reducing disability at short term follow up.** Data is grouped by the exercise training programme considered to be the experimental intervention with comparator interventions identified for each trial. Where trials report multiple treatment arms, each comparator is reported separately. Exercise training programmes compared to other eligible exercise training programmes are presented twice so that both treatment arms can be considered the experimental intervention. All values reported using Neck Disability Index unless otherwise indicated *A*–Progressive load in experimental arm; *B*–Fixed load in experimental arm; *C*–Patient specific functional scale; *D*–Northwick Park Neck Pain Questionnaire. Abbreviations: *SD*—Standard Deviation; *CI*—Confidence Interval; *AROM*—Active Range of Movement.

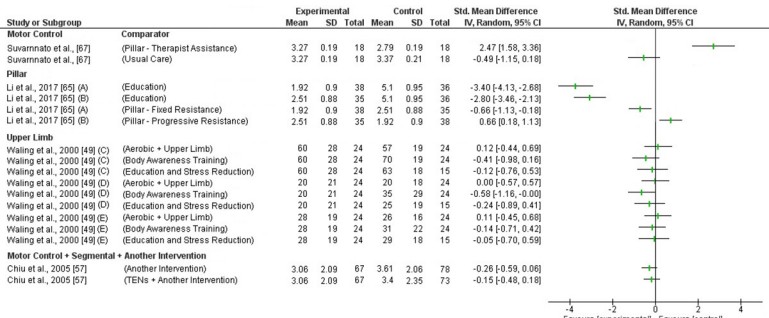

**Fig 6. Forest plot without a pooled estimate demonstrating the effectiveness of exercise training reducing pain at intermediate term follow up.** Data is grouped by the exercise training programme considered to be the experimental intervention with comparator interventions identified for each trial. Where trials report multiple treatment arms, each comparator is reported separately. Exercise training programmes compared to other eligible exercise training programmes are presented twice so that both treatment arms can be considered the experimental intervention. All values reported using Numeric Rating Scale/Visual Analogue Scale *A*–Progressive load in experimental arm; *B*–Fixed load in experimental arm; *C*–VAS at rest; *G*–VAS on activity; *C*–VAS in general; *D*–VAS at present; *E*–VAS at worst. Abbreviations: *SD*—Standard Deviation; *CI*—Confidence Interval.

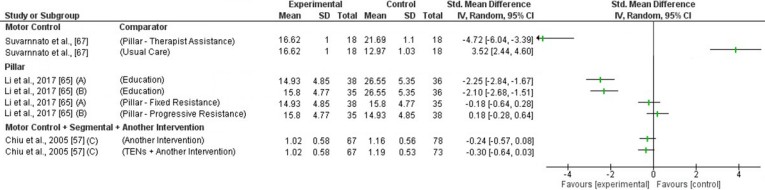

**Fig 7. Forest plot without a pooled estimate demonstrating the effectiveness of exercise training reducing disability at intermediate term follow up.** Data is grouped by the exercise training programme considered to be the experimental intervention with comparator interventions identified for each trial. Where trials report multiple treatment arms, each comparator is reported separately. Exercise training programmes compared to other eligible exercise training programmes are presented twice so that both treatment arms can be considered the experimental intervention. All values reported using Neck Disability Index unless otherwise indicated *A*–Progressive load in experimental arm; *B*–Fixed load in experimental arm; *C*–Northwick Park Neck Pain Questionnaire. Abbreviations: *SD* —Standard Deviation; *CI*—Confidence Interval.

(Present Pain: SMD -0.52; 95%ci -1.05 to 0.00) and body awareness training (Worst Pain: SMD -0.54; 95%CI -1.09 to 0.00; Present Pain: SMD -0.62; 95%CI -1.17 to -0.07) [49]. There was no effect compared to other exercise (general/worst pain) and body awareness training (general pain) when using different measures of pain [49]. One trial found consistent findings of no effect for intermediate term pain reduction but inconsistent findings for long term pain (Fig 6 and Fig 8) [49]. The trial found UL exercise to be less effective than education and stress reduction (present/general pain) but there was no effect compared to other exercise or body awareness training. Based on low level evidence (GRADE) UL exercises are effective reducing pain in the short term but not in the intermediate or long term.

**3.5.5 Motor control + pillar.** MC + pillar exercises was investigated in one trial (high RoB) showing a moderate effect reducing short term pain (SMD -0.66; 95%CI -1.23 to -0.09) and disability (SMD -1.23; 95%CI -1.84 to -0.63) compared to pillar exercise (Fig 4 and Fig 5) [71]. Based on low level evidence (GRADE) MC + pillar exercises are effective reducing short term pain and disability.

**3.5.6 Motor control + segmental.** MC + segmental exercises reducing short term pain was investigated in two trials (high RoB) with inconsistent findings (Fig 4). One trial found a very large effect compared to no treatment or other exercise [63]. Another trial found no effect when compared to no treatment [64]. Based on low level evidence (GRADE) MC + segmental exercises are effective reducing short term pain.

The same trials had inconsistent findings for MC + segmental exercises reducing short term disability (Fig 5). One trial found a very large effect compared to no treatment (SMD -4.30; 95%CI -5.34 to -3.26) but no effect when compared to other exercise [63]. The other trial found a large effect compared to no treatment (SMD -0.84; 95%CI -1.48 to -0.21) when

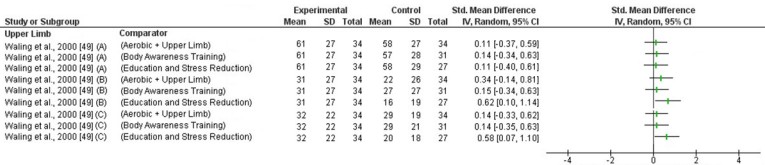

**Fig 8. Forest plot without a pooled estimate demonstrating the effectiveness of exercise training reducing pain at long term follow up.** Data is grouped by the exercise training programme considered to be the experimental intervention with comparator interventions identified for each trial. Where trials report multiple treatment arms, each comparator is reported separately. All values reported using Numeric Rating Scale/Visual Analogue Scale *A*–VAS in general; *B*–VAS at present; *C*–VAS at worst. Abbreviations: *SD*—Standard Deviation; *CI*—Confidence Interval.

measuring disability with Patient Specific Function Scale but no effect using the NDI [64]. Based on low level evidence (GRADE) MC + segmental exercises are not effective reducing short term disability.

**3.5.7 Motor control + segmental + pillar.** MC + segmental + pillar exercises was investigated in one trial [high RoB] and found a very large effect on pain (on activity SMD -2.71; 95% CI -3.87 to -1.55; at rest SMD -2.98; 95%CI -4.20 to -1.76) and disability (SMD -1.90; 95%CI -2.89 to -0.91) in the short term compared to other exercise (Fig 4 and Fig 5) [69]. Based on low level evidence (GRADE) MC + segmental + pillar exercises are effective reducing short term pain and disability.

**3.5.8 Motor control + segmental + another intervention.** MC + segmental exercise + another intervention reducing short term pain were investigated in two trials (high RoB) with inconsistent findings (Fig 4). One trial found a small to moderate effect when compared to another intervention or TENS [57]. One trial found a moderate to large effect reducing pain at best when compared to another intervention (pain best/now) or other exercise (pain best) [51]. The same trial found no effect reducing pain at worst. Based on moderate level evidence (GRADE) MC + segmental exercise + another intervention is effective reducing short term pain.

The same trials investigated MC + segmental exercise + another intervention reducing short term disability with inconsistent findings (Fig 5). One trial found a large effect compared to another intervention (SMD -1.12; 95%CI -1.79 to -0.44) or general AROM (SMD -1.03; 95%CI -1.69 to -0.37) [51]. The other trial found an effect compared to TENS (SMD -0.36; 95%CI -0.69 to -0.03) but no effect compared to another intervention [57]. Based on moderate level evidence (GRADE) MC + segmental exercise + another intervention is effective reducing short term disability.

Intermediate term pain and disability reduction was investigated in one trial (high RoB) showing no effect when compared to another intervention or TENS (Fig 6 and Fig 7) [57]. Based on low level evidence (GRADE) MC + segmental exercise + another intervention is not effective reducing intermediate term pain or disability.

**3.5.9 Segmental + upper limb.** Segmental + UL exercise reducing pain and disability was investigated in one trial (high RoB) showing no difference between two different dosages exercises at any time point (p > 0.05) [76]. Based on low level evidence (GRADE) different dosages of segmental + UL exercises does not change effectiveness of reducing short, intermediate, or long-term pain or disability.

**3.5.10 Other exercise training.** Based on very low to low level evidence (GRADE) the remaining ET programmes (Pillar + Another Intervention; Pillar + UL; Pillar + UL + Another Intervention; Segmental + UL; Segmental + UL + Another Intervention; UL + Segmental + MC + Another Intervention; XX + Upper Limb) are not effective reducing pain or disability (S8 Appendix).

## 3.6 Optimal dosage of different exercise training programmes

**3.6.1 Motor control.** Clinical heterogeneity (comparator/outcome measure) prevented comparison of MC dosage reducing immediate pain. Comparison of MC dosage reducing short term pain and disability was possible in two trials (1 high, 1 low RoB) with sufficient homogeneity and intervention reporting [59, 63]. The trials showed the direction of effect in reducing pain and disability moves in favour of MC exercises as the frequency increases (Table 4). Based on two trials increased dosage of MC exercise (frequency) potentially increases effectiveness.

**Table 4. Comparison of exercise dosage for Motor Control and Pillar exercise effectiveness in reducing pain/disability.**

| Exercise Training | Study | Comparator | Duration (weeks) | Frequency | | Intensity | | | | | | | Pain | Disability |
|---|---|---|---|---|---|---|---|---|---|---|---|---|---|---|
| | | | | Supervised (per week) | HEP (per day) | Reps | Sets | Load | Rep Rest (seconds) | Set Rest (seconds) | Tempo | | SMD [95%CI] | SMD [95% CI] |
| Motor Control Exercise | Borisut 2013[63] | Segmental | 12 | NA | x1 | 15 | 1 | 22-30mmHg | 10 | NA | 10 sec holds | | 0.29 [-0.27, 0.85] | -0.28 [-2.94, 2.38] |
| | Javanshir 2015[59] | Segmental | 10 | x3 | x3 | 10 | 1 | 22-30mmHg | 10 | NA | 10 sec holds | | -0.32 [-0.83, 0.19] | -3.81 [-8.69, 1.07] |
| Pillar Exercise | Li 2017* [65] | Education | 6 | x3 | NA | 8–12 | 1 | 70% maximal strength @ baseline | 5 | NA | ?? | | -2.37 [-2.99,-1.76] | -1.81 [-2.36,-1.25] |
| | Li 2017†[65] | Education | 6 | x3 | NA | 8–12 | 1 | Week 1–2–30% maximal strength @ baseline<br>Week 3–4–50% maximal strength @ week 2<br>Week 5–6–70% maximal strength @ week 4 | 5 | NA | ?? | | -2.89 [-3.55,-2.23] | -2.06 [-2.62,-1.49] |
| | Li 2017†[65] | Pillar (Fixed Resistance) | 6 | x3 | NA | 8–12 | 1 | Week 1–2–30% maximal strength @ baseline<br>Week 3–4–50% maximal strength @ week 2<br>Week 5–6–70% maximal strength @ week 4 | 5 | NA | ?? | | -0.74 [-1.21,-0.26] | -0.23 [-0.69, 0.23] |

* Fixed Resistance Pillar Exercise as Experimental Intervention,

† Progressive Resistance Pillar Exercise as Experimental Intervention,

?? Unable to extract data due to poor reporting. Abbreviations: *HEP*—Home Exercise Programme; *SMD*—Standardised Mean Difference; *CI*—Confidence Interval; *mmHg*—millimetres of Mercury; *NA*—Not Applicable

**3.6.2 Pillar.** One trial (high RoB) compared two dosages of pillar exercises in reducing short term pain and disability [65]. Starting with low load of exercise that is progressively increased is moderately more effective reducing pain than a fixed load (SMD -0.74; 95%CI: -1.21 to -0.26) but there is no difference in reducing disability (MD -1.15; 95%CI: -3.43 to 1.13)(Table 4). Based on one trial a low dosage (load) that is progressively increased potentially increases effectiveness reducing pain but not disability.

**3.6.3 Segmental.** Comparison of segmental exercise dosage reducing immediate pain was not possible as only one trial was found using a single dosage exercise [60].

**3.6.4 Motor control + segmental + another intervention.** Comparison of MC + segmental exercise + another intervention reducing short term pain and disability was not possible due to incomplete frequency and intensity reporting [51].

### 3.7 Additional analyses

Sensitivity analysis was not feasible due to clinical heterogeneity preventing meta-analysis.

## 4 Discussion

### 4.1 Summary of evidence

This the first systematic review to investigate the effectiveness of ET programmes categorised by their intended effect on spinal function on CNSNP and/or disability and whether dosage alters outcomes. Fifteen ET programmes from 26 trials were evaluated. Clinical heterogeneity, poor intervention reporting, low sample sizes and high RoB limits our understanding of ET effectiveness and the influence of dosage.

**4.1.1 Effectiveness.** No ET programmes were found to be effective at immediate or long term follow up. Pillar exercises were found to be effective at intermediate term compared to no treatment (low level evidence). Most trials investigated short term effectiveness of which multiple ET programmes encompassing exercises for various spinal functions demonstrated benefits. Effectiveness was predominantly found in ET packages containing MC exercises, suggesting retraining co-ordination or sequential control of neck movement is a critical component to exercise for CNSNP. Interestingly, when MC exercises were used alone the effectiveness was unclear, however benefits are maximised when combined with other ET. Based on effect size and overall quality of evidence the best outcomes were achieved when MC exercises were combined with segmental exercises [MC + segmental exercises (low level evidence); MC + segmental exercises + another intervention (moderate level evidence) and MC + segmental + pillar exercises (low level evidence)].

This is the first systematic review identifying combinations of MC + segmental exercises as the most effective ET for patient reported outcomes. One explanation for this is that MC exercises improve neuromuscular function of the deeper cervical muscles and segmental exercises improve strength, endurance and fatiguability of the superficial cervical muscles [18, 19, 79, 80]. Combining exercises results in multiple improvements in neuromuscular and spinal function impairments common in CNSNP [21, 81].

**4.1.2 Dosage.** There is conflicting evidence whether higher dosages of exercise results in greater reductions in musculoskeletal pain or disability [82–91]. This is largely due to the limited number of trials directly comparing two identical ET programmes at different dosages. A meta-analysis of an existing data set from a Cochrane systematic review on exercise for chronic neck pain found a positive correlation between exercise duration (in weeks) and a reduction in neck pain [91]. It is not clear whether this applies to the ET programmes considered in our review as the analysis was completed on studies using "gymnastics" "qigong" "flexibility" exercises in addition to "strength" exercises. Secondary analysis of neck pain trial data suggests that higher dosages through increased adherence or greater sets and repetitions seems to have greater benefits in neck pain [25, 26, 53]. However, none of these studies used ET programmes that consisted of MC + segmental exercises. MC exercises focus on neuromuscular relearning and quality of movement typically using lighter loads whereas segmental exercises aiming to increase the force production of muscles often require higher loads [20]. Manipulation of exercise variables and dosage are key in classifying exercise as motor control or segmental, meaning precision prescription is required to ensure the desired effect on spinal and neuromuscular function is achieved. This evidence synthesis was unable to provide guidance as to exercise and dosage variables required to maximise the effectiveness of MC + segmental exercise. Although the effectiveness of MC exercises alone potentially improves as frequency increases this was based on limited data and we were unable to investigate segmental exercise dosage.

Pillar exercise effectiveness improves if load is progressively increased, but it is unknown whether this also applies to segmental exercise.

**4.1.3 Evidence quality.**   Clinical heterogeneity and intervention reporting limited MC + segmental dosage analysis. Exercise variables (delivery, equipment etc) differed between trials meaning differences in effect could be due to these components rather than dosage differences. If homogeneity existed, intervention reporting limited dosage data extraction further preventing comparison. Reporting checklists TIDieR (published 2014) [38] and CERT (published 2016) aid intervention reporting but only 1 MC + segmental exercise trial was reported after 2014 [51]. As a result, trial authors did not use these tools to provide intervention clarity. Consequently, different MC + segmental exercise variables and poor intervention reporting are problematic for researchers wanting to perform further studies or meta-analysis [92]. More importantly it provides little guidance for clinicians to deliver MC + segmental exercise effectively for CNSNP patients. [38]

Confidence in findings for packages of MC + segmental exercise is reduced due to high RoB (participant blinding/selective outcome reporting) and imprecision meaning the true effect of MC + segmental exercise maybe different to that reported [93–95]. An adequately powered, low risk of bias trial would improve confidence in findings. Furthermore, future trials must evaluate long term effectiveness due to the recurring nature of neck pain [7, 8].

## 4.2 Comparison with other systematic reviews

Previous systematic reviews demonstrating exercise effectiveness base findings on trials that we assessed as using UL or UL + Pillar exercises [10, 11]. The design of this review provides advantages over other reviews offering evidence previously not synthesised. Firstly, any exercise comparator was eligible resulting in different included trials. Secondly, to improve participant homogeneity, participants were required to have symptoms ≥3 months and therefore excluded trials [cited in previous reviews] using participant eligibility of pain >30 days within the last year [15, 96–100]. Therefore, although we found inconsistent evidence of UL exercise short term effectiveness, we found consistent evidence that UL are not ineffective in the long term and UL + pillar exercises are not effective at any time point. Unlike another review supporting MC exercises, we found inconsistent evidence that MC exercises alone are not effective [101]. The meta-analysed results from Martin-Gomez, Sestelo-Diaz (101) should be treated with caution due to substantial statistical heterogeneity ($I^2$ = 66–67%).

## 4.3 Implications

This systematic review has important implications for clinical practice. CNSNP treatment should include combinations of submaximal effort exercises for the deep cervical muscles to relearn or improve neuromuscular movement patterns (motor control exercises) and exercises to improve the ability of the larger superficial cervical muscles to produce force (segmental exercises). Optimal motor control and segmental exercise dosage is unclear due to the significant clinical heterogeneity between trials. Future research should gain consensus on key exercise and dosage variables that can be explored further within a complex intervention framework. [102] Long-term the effectiveness and optimal dosage of MC + segmental exercise needs evaluation through an adequately powered, low RoB clinical trial.

## 4.4 Strengths

This is the first systematic review focusing on effectiveness and optimal dosage of subgroups of ET based on the intended effect on spinal function in CNSNP. This review employed a

rigorous methodology and was conducted according to a published protocol and reported in line with PRISMA guidance.

### 4.5 Limitations

Excluding 4 non-English and 16 inaccessible studies could be a limitation; however, the increased studies may have contributed to further clinical heterogeneity. Poor intervention reporting may have led to inaccurate ET classification. Using a different exercise classification tool may lead to different conclusions. The quality of the included studies was reduced due to heterogeneity of outcome measures, low sample sizes and high risk of bias of the included studies reduced evidence quality.

## 5 Conclusions

Low to moderate evidence supports the effectiveness of ET packages that include MC + segmental exercises reducing patient reported outcomes at short term follow up. This is based on high RoB trials utilising different exercise variables. The long-term effectiveness of MC + segmental exercise has not been evaluated. MC + segmental exercise variables including dosage need to be defined and investigated in an adequately powered low RoB clinical trial with long term follow up.

## Supporting information

**S1 Appendix. PRISMA 2009 checklist.**
(PDF)

**S2 Appendix. Database searches.**
(PDF)

**S3 Appendix. Data extraction form.**
(PDF)

**S4 Appendix. Risk of bias.**
(PDF)

**S5 Appendix. GRADE.**
(PDF)

**S6 Appendix. Excluded studies.**
(PDF)

**S7 Appendix. Table of characteristics.**
(PDF)

**S8 Appendix. Mean, sd and smd for all exercise training.**
(PDF)

## Acknowledgments

Academic support provided from University of Birmingham and the Centre of Precision Rehabilitation for Spinal Pain including patient users involved in the interpretation of the results. The authors would like to thank Simon Spencer for his advice in exercise classification.

## Author Contributions

**Conceptualization:** Jonathan Price, Alison Rushton, Nicola R Heneghan.

**Formal analysis:** Jonathan Price, Isaak Tyros, Vasileios Tyros.

**Investigation:** Jonathan Price, Isaak Tyros, Vasileios Tyros.

**Methodology:** Jonathan Price, Alison Rushton, Isaak Tyros, Nicola R Heneghan.

**Project administration:** Jonathan Price.

**Resources:** Jonathan Price.

**Supervision:** Alison Rushton, Nicola R Heneghan.

**Validation:** Jonathan Price, Alison Rushton, Isaak Tyros, Vasileios Tyros, Nicola R Heneghan.

**Visualization:** Jonathan Price.

**Writing – original draft:** Jonathan Price, Alison Rushton, Nicola R Heneghan.

**Writing – review & editing:** Jonathan Price, Alison Rushton, Isaak Tyros, Vasileios Tyros, Nicola R Heneghan.

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
