## [Decision Letter · Decision Letter 0]

22 Apr 2020

PONE-D-20-07090

Effectiveness and optimal dosage of exercise training for chronic non-specific neck pain: a systematic review with a qualitative synthesis

PLOS ONE

Dear Dr Heneghan,

Thank you for submitting your manuscript to PLOS ONE. After careful consideration, we feel that it has merit but does not fully meet PLOS ONE’s publication criteria as it currently stands. Therefore, we invite you to submit a revised version of the manuscript that addresses the points raised during the review process.

We would appreciate receiving your revised manuscript by one month. To enhance the reproducibility of your results, we recommend that if applicable you deposit your laboratory protocols in protocols.io, where a protocol can be assigned its own identifier (DOI) such that it can be cited independently in the future. For instructions see: http://journals.plos.org/plosone/s/submission-guidelines#loc-laboratory-protocols

We look forward to receiving your revised manuscript.

Kind regards,

Jean-Philippe Regnaux, Ph.D, PT

Academic Editor

PLOS ONE

Journal Requirements:

Additional Editor Comments :

Dear Authors, thank you for having given us the opportunity to read you work. Experts and I have carefully read your submission. We found that your manuscript has the potential to be published in PLOS ONE. The reviewers have adressed some comments. So i would like to invite you answer the reviewers' comments and revise your manuscript. Sincerely.

Reviewers' comments:

Reviewer's Responses to Questions

**Comments to the Author**

1. Is the manuscript technically sound, and do the data support the conclusions?

Reviewer #1: Yes

Reviewer #2: Yes

Reviewer #3: Partly

2. Has the statistical analysis been performed appropriately and rigorously? 

Reviewer #1: Yes

Reviewer #2: Yes

Reviewer #3: I Don't Know

3. Have the authors made all data underlying the findings in their manuscript fully available?

Reviewer #1: Yes

Reviewer #2: Yes

Reviewer #3: Yes

4. Is the manuscript presented in an intelligible fashion and written in standard English?

Reviewer #1: Yes

Reviewer #2: Yes

Reviewer #3: Yes

5. Review Comments to the Author

Reviewer #1: Summary: This systematic review addresses a critical unknown in exercise training for pain. The most of effective dose of exercise (type, intensity, duration, frequency) is unclear due to a lack of individual clinical trials evaluating dose. In this manuscript, the authors use a systematic review approach to determine whether such data may exist across studies. The protocol for this review (with meta analysis or systematic review option) was published in February 2019 (BMJ Open). The study sourced data from 3990 citations (26 selected). Low sample sizes and high risk of bias prevented a meta-analytic approach for analysis. Overall methodology was strong. Results are generally clearly described. Table 1 is informative in the analysis of the types of ET considered a priori versus what was found in studies. This information will be important for the next iteration of this study completed in the future. Overall, the final conclusions of the study are muted based on high levels of RoB and clinical heterogeneity. Some additional interpretation of how the unanswered questions could be addressed in future clinical trials would be welcomed. Nonetheless, this is a well-executed study that makes an important contribution on the neck pain therapy field.

Major:

• Introduction Page 5, Lines 107-110 - Author states “clinical guidelines do not provide dosage recommendations...” What about ACSM recommendations for exercise (150 min per week for moderate intensity exercise for healthy populations)?

• Results - Overall, there needs to be more synthesis in the results section from a hypothesis standpoint that would leave the readers with more specific clinical research questions to answer. The authors are careful to not speculate on clinical practice but I think speculation on clinical research that could help inform the gaps identified in this review would be valuable to assess.

• Figure legends are too brief. Please add additional detail in these legends to ease burden on reader. For example, in Figure 2, there are many more than 26 individual exercises. It should be clear in the legend that some trials included multiple types of ET.

• Regarding exercise dose, was their any information on total time (per week) given in studies. Please comment in results on time per week which can be correlated with frequency per week but is not necessarily so).

Minor:

• Overall, the writing could be more concise and have more clarity throughout.

• Introduction Page 4, Line 83-86 - this sentence is confusing and should be broken up/re-written for clarity.

• Introduction Page 4, Line 84 - Authors state that “The effects of exercise are poorly understood as despite short term benefits…” I don’t think this is true can be written differently.

• Please include inception date of study in methods when describing search date range

• Methods Page 6, Eligibility Criteria - Authors state “Eligibility criteria were developed by scoping searches and PICOS.” What are the eligibility criteria for the studies? This is very vague and should be listed out in methods, not just the supplemental figures.

• Methods It is not clear where this “GRADE” system comes from. Was this developed by the authors? More detail should be included for why this was used and how it was developed.

• Please clarify sentence in methods line 194-195. “Clinically importance differences were established a priori as a mean difference of 1/10 VAS and 5/50 NDI [10, 42].” Does this mean “a mean difference > 1 on a 10-point VAS scale or a score > 5 on the 50-total score NDI scale”?

• Methods Line 202-203 - Clarify “Mean scores using a reverse scale were multiplied by -1 [35].” Does that mean that positive scores are indicative of positive pain effect (good efficacy) or negative pain effect (poor efficacy).

• Another study that assesses chronic neck pain using strengthening exercises was omitted and should be consider for inclusion/exclusion in analysis. (See Beer et al. 2012).

• Results Figure 1 – Please indicate exclusion criteria “Protocol”.

• Results line 231 – Consider moving line 231 above sentence beginning 229 “Multiple reports…” and add “from 33 citations” after “…for 26 trials” in line 231.

• Results Page 15, Lines 299-308 - It is unclear which results the authors are referring to (pain or disability or combined)?

• Results line 271 – Please provide list/table of interventions used per trial.

• Results - Page 16, Line 328 - change “effectiveness” to effective.

• Results - Page 17, Line 356 - authors include statement “based on low level evidence (GRADE)..” This is done again at the beginning of the results section but is not repeated for similarly made statements throughout. Either include “(GRADE)” with each statement or only with the first statement of the first section of results.

• Results Page 17, Line 352 - Authors state “The effect on disability was clinically important,” please include criteria in parenthesis afterwards.

• Results Page 18, Lines 360-363 - Please include stats.

• Results Page 19, Line 405 - Please spell out AROM.

• Results Page 23, Line 439 - Please change “reducing” to reduced.

• Discussion Page 27, Limitations - Please add additional limitations (e.g. low sample size, high RoB, different exercise variables)

• Discussion – With regard to previous analyses and context authors should consider Geneen et al. 2017 for Cochrane exercise meta-meta analysis of literature and Polaski et al. 2019 (Plos One) for meta-analysis of these data for pain and exercise dosage. (see specifically Neck Pain)

Reviewer #2: It was with great interest that I read this study. I think the authors have written an excellent manuscript. The findings weren't what I hoped and I really like how the authors were able to make it clear what studies need to be done to provide sufficient information for chronic neck pain. I have a few suggestions that I think may help make this quality manuscript even better.

Minor concerns.

1.2 While the authors provide a purpose I think that adding a hypothesis is pertinent. It really limits the reader if they don't know what you are thinking.

Page 9, line 184. I'd define RoB here again just to assist the reader.

Page 10, line 222. If the patients recommended exercise, then quotations would be appropriate.

Reviewer #3: This systematic review with a qualitative synthesis aimed to evaluate the current evidence on the effectiveness of different exercise training programmes on reducing chronic non-specific neck pain and disability, and whether dosage affects outcome. This is a great topic that is clearly of interest and relevance to clinicians and to the readers of the PlosOne journal. I have a few main and minor comments/suggestions that I have written in their order of appearance in the manuscript.

Suggested revisions:

Main comments

Comment 1: Title: This is not my expertise but I think the aim of a qualitative evidence synthesis is to summarise the evidence from qualitative studies. I do not think you have included qualitative studies in this systematic review, where you could gather information on qualitative data to synthetise the evidence. All data included in this review are quantitative data. I would suggest you to remove qualitative synthesis from the title and maybe include “a systematic review with a narrative synthesis”.

Comment 2. Methods: Summary Measures - Categorisation of the various “Exercise Training” as experimental intervention and control. While this is understandably a difficult thing to do to everyone’s satisfaction, I am not sure that the current way to present the study Synthesis of Results is clear. A little more clarity may help. I appreciate you will be more familiar with these studies, so if this can be more clear, it would be great:

- For instance, for Borisut et al 2013 ‘Motor Control intervention’ is listed as experimental intervention in fig 5 (disability, short-term), whereas ‘Motor Control + Segmental intervention’ is listed as Comparator/Control with Std. Mean difference (SMD -0.31, 95% CI -0.87 to 0.25). In the same fig 5, but under the Motor Control + Segmental heading, for Borisut et al 2013 ‘Motor Control + Segmental’ intervention is listed as experimental intervention, whereas ‘Motor Control’ is listed as Comparator/Control with Std. Mean difference (SMD 0.31, 95% CI -0.25 to 0.87). I find it hard to see how these 2 identical intervention contrast are summarised as both interventions are labelled as an experimental intervention in one comparison and a Comparator/control intervention in another. I suggest you revise figure 5 and similarly fig 4, please and if possible restructure your summary table. One suggestion is that, even if you could not pooled estimates in a meta-analysis, you could have the headings structured as intervention contrasts such as “Motor Control” vs “No Treatment” and follow this structure of summary for both outcomes of Pain Intensity and Disability. I believe that way it would be more clear for readers to follow your study narrative synthesis in both figures and in the result section of the manuscript.

- Furthermore, I notice you have only presented data in fig 4 (pain, short-term follow-up) and fig 5 (disability, short-term follow-up). It would be great, to see the summary synthesis for all outcomes and time-points data extracted, not only for short-term.

Comment 3. Methods: Summary Measures – Please provide a reference and further clarification on the analysis approach used to calculate SMD and MD: change score or outcome analysis? In some studies it seem you have used outcome analysis (e.g. Borisut et al. 2013; Chung et al. 2018), and others you have used change scores (e.g. Bobos et al. 2016). Also, it would be great if you could mention the reason for the selection of approach and the software used to calculate SMD and MD with respective uncertainty interval.

Comment 4. Methods: Summary Measures – On line 195 you have stated that “Clinically importance differences were established a priori as a mean difference of 1/10 VAS and 5/50 NDI”. You have used Standard Mean Difference to estimate the effect for a few of the intervention contrast in this review. I was wondering if you could clarify that please? Furthermore, I was wondering what have you established as a clinically important difference for other outcomes such as Northwick Park Neck Pain Questionnaire and Neck and Shoulder Pain and Disability Index?

Comment 5. Methods: Summary Measures – Following the previous comment, I was wondering if you could please clarify the use of Mean Difference (MD) for some studies and Standardised Mean Difference (SMD) for others.

Comment 6. Methods: Synthesis of results – To rate the quality of evidence you have used the Grading of Recommendations, Assessment, Development and Evaluation (GRADE) system. In S5 Appendix I can see you have pooled studies by outcome time-points (Immediate-term, Short-term, Intermediate-term, and Long-term). The overall certainty in the evidence should be assessed for each review outcome and for all studies pooled in the meta-analysis (pain-intensity and disability in this review). In this review it was not possible to pooled RCTs in a meta-analysis, so the GRADE should be assessed at individual RCT level for each intervention contrast and respective outcomes (e.g. “Motor Control vs No treatment” for both pain-intensity and disability). I suggest you review the quality of evidence assessment and modify S5 Appendix table accordingly.

Comment 7. Results: Effectiveness of different exercise training programmes – I would suggest, if it is possible, to organise the narrative synthesis in results section by intervention contrast. It is just a suggestion as I believe it will help and probably be more clear for readers to see the effectiveness of the experimental intervention against comparator/control groups. I found it hard to follow.

For instance, you could have Motor Control vs No treatment:

- 3.5.1 Motor Control vs No treatment

The short-term effect of Motor control (MC) compared to No treatment on pain-intensity and disability was investigated in one trial (Borisul et al., 2013). There is xxx-quality evidence that ET involving MC is effective for pain-intensity (MD -1.17, 95% CI -1.78 to -0.57), and xxx-quality evidence that ET involving MC is effective for disability (MD -3.84, 95% CI -4.80 to -2.88) when compared to No treatment control.

Minor comments

Comment 8. Abstract: Objective – I suggest re-word objective to: “To synthesise the current evidence on the effectiveness of different ET programmes to reduce CNSNP and associated disability, and whether dosage affects outcomes.”

Comment 9. Abstract: Results – I suggest you, whenever possible, to present the point estimate with respective uncertainty interval. Same for the result section.

Comment 10. Abstract: Results – Would be more clear if you could replace the term “Multiple” (in line 62) for the combination of ET programmes that reduced the outcome of pain-intensity and disability in the short-term. At the moment it is a bit confuse what multiple means.

Comment 11. Introduction: Please provide, a reference or further explanation/definition for what constituted an "Exercise Training (ET)" for this review.

Comment 12. Methods: Outcome Measures – I suggest to add the word “neck”. So the sentence reads: “Any patient reported measure of neck pain [e.g. Visual Analogue Scale (VAS)) and/or neck functional disability (e.g. Neck Disability Index, (NDI)].”

Comment 13. Methods: Synthesis of results – Maybe use the word narrative instead of quality in the following sentence on line 210. “Where meta-analysis was not possible, narrative synthesis provided summaries of the evidence.”

6. PLOS authors have the option to publish the peer review history of their article (what does this mean?). If published, this will include your full peer review and any attached files.

Reviewer #1: No

Reviewer #2: No

Reviewer #3: Yes: Tarcisio Folly de Campos

---

## [Author Response · Author response to Decision Letter 0]

16 May 2020

Reviewer Comments Authors Response

Reviewer 1

Summary 

This systematic review addresses a critical unknown in exercise training for pain. The most of effective dose of exercise (type, intensity, duration, frequency) is unclear due to a lack of individual clinical trials evaluating dose. In this manuscript, the authors use a systematic review approach to determine whether such data may exist across studies. The protocol for this review (with meta analysis or systematic review option) was published in February 2019 (BMJ Open). The study sourced data from 3990 citations (26 selected). Low sample sizes and high risk of bias prevented a meta-analytic approach for analysis. Overall methodology was strong. Results are generally clearly described. Table 1 is informative in the analysis of the types of ET considered a priori versus what was found in studies. This information will be important for the next iteration of this study completed in the future. Overall, the final conclusions of the study are muted based on high levels of RoB and clinical heterogeneity. Some additional interpretation of how the unanswered questions could be addressed in future clinical trials would be welcomed. Nonetheless, this is a well-executed study that makes an important contribution on the neck pain therapy field. Thank you for your constructive and positive comments on our submission. 

Major 

• Introduction Page 5, Lines 107-110 - Author states “clinical guidelines do not provide dosage recommendations...” What about ACSM recommendations for exercise (150 min per week for moderate intensity exercise for healthy populations)? Thank you for raising this point. While we are familiar with the ACSM guidelines our statement was in reference to neck pain clinical guidelines. We have now updated the text to make this point clearer.

• Results - Overall, there needs to be more synthesis in the results section from a hypothesis standpoint that would leave the readers with more specific clinical research questions to answer. The authors are careful to not speculate on clinical practice but I think speculation on clinical research that could help inform the gaps identified in this review would be valuable to assess. We agree with the reviewers that alternative approaches to evidence synthesis could have provided more clarity. In a very early draft of the manuscript the results were synthesised by comparing ET programmes to groups of comparator e.g motor control v no treatment; motor control v usual care etc

However due to comparator heterogeneity assessing the evidence in this way would not allow us to synthesize evidence which was the primary aim. 

E.g for Motor Control ET Programmes the comparisons would have been

Motor Control vs No Treatment (1 trial)

Motor Control vs Usual Care (1 trial)

Motor Control vs AROM (1 trial)

Motor Control vs Segmental (2 trials)

Motor Control vs Pillar (4 trials)

Motor Control vs Motor Control & Segmental (1 trial)

Motor Control vs Proprioceptive Training (1 trial) 

To clarify this point to the reader we have added the following statement under “3.4 Results of individual studies and synthesis of results”

“Comparator heterogeneity also prevented the synthesis or analysis of evidence quality by contrasting each ET programme to subgroups of comparator interventions (e.g Motor Control vs No Treatment). Therefore, the effectiveness of each ET programme has been narratively described against all reported comparators.”

To respond to the reviewers comments regarding suggestions for future clinical research. We have updated section 4.3 Implications, to include the following

“Optimal motor control and segmental exercise dosage is unclear due to the significant clinical heterogeneity between trials. Future research should gain consensus on key exercise and dosage variables that can be explored further within a complex intervention framework.[102]”

• Figure legends are too brief. Please add additional detail in these legends to ease burden on reader. For example, in Figure 2, there are many more than 26 individual exercises. It should be clear in the legend that some trials included multiple types of ET. Thank you. We have updated our figures to provide more detail.

To provide further clarify around Figure 2 we have removed this figure and added an extra table with a description of interventions for each study. We had also included each of the exercise training programmes referencing respective trials in Section 3.2.3

• Regarding exercise dose, was their any information on total time (per week) given in studies. Please comment in results on time per week which can be correlated with frequency per week but is not necessarily so). Time spent exercises is an interesting point. Only 53% of trials reported enough information to calculate treatment time per week (ranging from 45 mins to 270 mins). However, this time includes non-exercise training interventions such as stretching, education, manual therapy. We have updated the results section to reflect this point

“Supervised treatment session time including non-ET interventions, ranged from 45 to 270 mins per week. Poor intervention reporting limited analysis of time spent completing ET at home or during supervised sessions”

Minor 

• Overall, the writing could be more concise and have more clarity throughout. We have revised the manuscript and added clarity in certain areas the reviewers commented on. 

• Introduction Page 4, Line 83-86 - this sentence is confusing and should be broken up/re-written for clarity. Thank you for identifying this. This has now been re-written 

“Despite short term benefits of exercise, long term effectiveness is unclear as 70% of individuals will develop recurring or persistent chronic non-specific neck pain (CNSNP)”

• Introduction Page 4, Line 84 - Authors state that “The effects of exercise are poorly understood as despite short term benefits…” I don’t think this is true can be written differently. We have rewritten this statement as per the previous comment.

• Please include inception date of study in methods when describing search date range We would like to direct the reviewer to the following points

• 2.2.6 where we state there will be no date restrictions

• 2.3. states inception to 6/1/2020

• Methods Page 6, Eligibility Criteria - Authors state “Eligibility criteria were developed by scoping searches and PICOS.” What are the eligibility criteria for the studies? This is very vague and should be listed out in methods, not just the supplemental figures. Our scoping searches served only to inform our search strategy which is detailed in full of 2.2.1 to 2.2.6. 

 We hope clarification here resolves this to the satisfaction of the reviewer.

• Methods It is not clear where this “GRADE” system comes from. Was this developed by the authors? More detail should be included for why this was used and how it was developed. GRADE is the framework recommended by Cochrane to aid in developing and presenting summaries of quality of evidence for clinical practice. We have included further references which provides full details of how GRADE was developed and how to use the framework.

• Please clarify sentence in methods line 194-195. “Clinically importance differences were established a priori as a mean difference of 1/10 VAS and 5/50 NDI [10, 42].” Does this mean “a mean difference > 1 on a 10-point VAS scale or a score > 5 on the 50-total score NDI scale”? Thank you for suggesting clarity over this point. A clinically important difference in pain was measured as a score of greater than 1 on a 10 point VAS scale. A clinically important difference in disability was measured as a score of greater than 5 on a 50 point NDI scale. We have updated the manuscript to provide more clarity

“Clinically importance differences were established a priori as a mean difference of >1/10 VAS for pain and >5/50 NDI for disability [10, 42]” 

• Methods Line 202-203 - Clarify “Mean scores using a reverse scale were multiplied by -1 [35].” Does that mean that positive scores are indicative of positive pain effect (good efficacy) or negative pain effect (poor efficacy). Thank you. Higher scores on some scales reflect a ‘better’ outcome and other scales use lower scores reflect a ‘better’ outcome. To ensure that all scales point in the scale direction the mean value of one set of scales needs to be multiplied by -1. 

The text has been updated to clarify this point.

“To ensure lower scores reflect a “better” outcome for all scales, mean scores for any outcome measures using a reverse scale (i.e where a lower score reflects a “worse” outcome) were multiplied by -1”

• Another study that assesses chronic neck pain using strengthening exercises was omitted and should be consider for inclusion/exclusion in analysis. (See Beer et al. 2012). Thank you. As per our protocol we excluded pilot or feasibility studies. As Beer et al., 2012 was reported as a “preliminary study” it failed to meet our inclusion criteria and was excluded at title and abstract screening. On further evaluation of the full text it also appears the population did not meet our inclusion criteria as it is not clear that WAD patients were excluded.

We have updated the inclusion criteria to reflect the protocol more accurately; exclusion of pilot or feasibility studies.

• Results Figure 1 – Please indicate exclusion criteria “Protocol”. Protocols for trials not yet completed were excluded at full text stage. We have updated section 2.2.6 to reflect this.

“Trials not written in English and protocols for trials not yet completed were excluded at full text and reported within the PRISMA flow diagram.”

• Results line 231 – Consider moving line 231 above sentence beginning 229 “Multiple reports…” and add “from 33 citations” after “…for 26 trials” in line 231. Thank you for this suggestion however following further discussion re flow we feel this works best as it is currently presented.

• Results Page 15, Lines 299-308 - It is unclear which results the authors are referring to (pain or disability or combined)? Thank you for identifying the lack of clarity here. We have updated this section to identify that it was both pain and disability

“Five trials (4 high RoB, 1 low RoB) found a moderate to very large effect on pain and disability compared to no treatment, usual care, general active range of movement (AROM) or pillar exercise [13, 63, 67, 70, 75]. Four trials (3 high RoB, 1 low RoB) found no effect on pain and disability compared to other ET and proprioceptive training [59, 63, 72, 74].”

• Results line 271 – Please provide list/table of interventions used per trial. Full intervention data can be found in S7 Appendix. To make it easier for the reader we have added another table that outlines brief intervention and dosage information for each trial found within the main text. 

• Results - Page 16, Line 328 - change “effectiveness” to effective. Thank you. We have updated this error

• Results - Page 17, Line 356 - authors include statement “based on low level evidence (GRADE)..” This is done again at the beginning of the results section but is not repeated for similarly made statements throughout. Either include “(GRADE)” with each statement or only with the first statement of the first section of results. Thank you for identifying this. We have updated the manuscript to identify the use of GRADE throughout.

• Results Page 17, Line 352 - Authors state “The effect on disability was clinically important,” please include criteria in parenthesis afterwards. Due to consider heterogeneity and to make it easier for the reader we have used SMD throughout and added the following under section 3.4

“Outcome measures heterogeneity limited summary measures to standardised mean differences plus 95% CI’s.”

In light of this we are unable to make the statement regarding clinical importance and has therefore been removed.

• Results Page 18, Lines 360-363 - Please include stats. Thank you. The text has now been updated to include the SMD data

• Results Page 19, Line 405 - Please spell out AROM. Thank you. AROM was written out in full in Section 3.5.1 and followed by an abbreviation for us to use throughout the rest of the text.

• Results Page 23, Line 439 - Please change “reducing” to reduced. Thank you. This has now been updated to read 

“One trial (high RoB) compared two dosages of pillar exercises in reducing short term pain and disability [65]. “

• Discussion Page 27, Limitations - Please add additional limitations (e.g. low sample size, high RoB, different exercise variables) Thank you. We have added this into Section 4.5 Limitations

• Discussion – With regard to previous analyses and context authors should consider Geneen et al. 2017 for Cochrane exercise meta-meta analysis of literature and Polaski et al. 2019 (Plos One) for meta-analysis of these data for pain and exercise dosage. (see specifically Neck Pain) Thank you. This is a series of work we are familiar with however initially excluded it from our discussion as the meta-analysed data was not from studies using exercise types similar to our review. We have now updated our discussion to include the following

“A meta-analysis of an existing data set from a Cochrane systematic review on exercise for chronic neck pain found a positive correlation between exercise duration (in weeks) and a reduction in neck pain.[91] It is not clear whether this applies to the ET programmes considered in our review as the analysis was completed on studies using “gymnastics” “qigong” “flexibility” exercises in addition to “strength” exercises.”

Reviewer 2 

Summary 

Reviewer #2: It was with great interest that I read this study. I think the authors have written an excellent manuscript. The findings weren't what I hoped and I really like how the authors were able to make it clear what studies need to be done to provide sufficient information for chronic neck pain. I have a few suggestions that I think may help make this quality manuscript even better Thank you very much for your very complimentary comments on our manuscript. 

Minor 

1.2 While the authors provide a purpose I think that adding a hypothesis is pertinent. It really limits the reader if they don't know what you are thinking. Thank you. We have added the following in Section 1.2 

“There are two hypotheses to this systematic review:

1. 1. Exercise training programmes categorised by their intended effect on spinal function have different effects on chronic non-specific neck pain and disability

2. 2. Exercise training programmes of different dosages have different effects on chronic non-specific neck pain and disability”

Page 9, line 184. I'd define RoB here again just to assist the reader. Thank you. We have updated the text accordingly

Page 10, line 222. If the patients recommended exercise, then quotations would be appropriate. Thank you for identifying this. We have added quotation marks around exercise training to highlight this point

Reviewer 3 

Summary 

This systematic review with a qualitative synthesis aimed to evaluate the current evidence on the effectiveness of different exercise training programmes on reducing chronic non-specific neck pain and disability, and whether dosage affects outcome. This is a great topic that is clearly of interest and relevance to clinicians and to the readers of the PlosOne journal. I have a few main and minor comments/suggestions that I have written in their order of appearance in the manuscript. Thank you very much for your very positive comments on our manuscript.

Major 

Title: This is not my expertise but I think the aim of a qualitative evidence synthesis is to summarise the evidence from qualitative studies. I do not think you have included qualitative studies in this systematic review, where you could gather information on qualitative data to synthetise the evidence. All data included in this review are quantitative data. I would suggest you to remove qualitative synthesis from the title and maybe include “a systematic review with a narrative synthesis”. Thank you for raising an interesting point. We had originally used the term narrative synthesis in the original version of our protocol but were recommended by the reviewers to use the term “Qualitative Synthesis”. In order to maintain consistency with the protocol we used the term qualitative synthesis in this paper.

Our preference is to use the term narrative synthesis and as such have updated the manuscript to reflect this. 

Methods: Summary Measures - Categorisation of the various “Exercise Training” as experimental intervention and control. While this is understandably a difficult thing to do to everyone’s satisfaction, I am not sure that the current way to present the study Synthesis of Results is clear. A little more clarity may help. I appreciate you will be more familiar with these studies, so if this can be more clear, it would be great:

- For instance, for Borisut et al 2013 ‘Motor Control intervention’ is listed as experimental intervention in fig 5 (disability, short-term), whereas ‘Motor Control + Segmental intervention’ is listed as Comparator/Control with Std. Mean difference (SMD -0.31, 95% CI -0.87 to 0.25). In the same fig 5, but under the Motor Control + Segmental heading, for Borisut et al 2013 ‘Motor Control + Segmental’ intervention is listed as experimental intervention, whereas ‘Motor Control’ is listed as Comparator/Control with Std. Mean difference (SMD 0.31, 95% CI -0.25 to 0.87). I find it hard to see how these 2 identical intervention contrast are summarised as both interventions are labelled as an experimental intervention in one comparison and a Comparator/control intervention in another. I suggest you revise figure 5 and similarly fig 4, please and if possible restructure your summary table. One suggestion is that, even if you could not pooled estimates in a meta-analysis, you could have the headings structured as intervention contrasts such as “Motor Control” vs “No Treatment” and follow this structure of summary for both outcomes of Pain Intensity and Disability. I believe that way it would be more clear for readers to follow your study narrative synthesis in both figures and in the result section of the manuscript.

 The reviewer makes a good point here regarding the Synthesis of Results. Firstly, we want to directly comment on the example provided. The Borisut et al 2013 paper had 4 treatment arms, 1. Motor control 2. Segmental 3. Motor control + Segmental 4. No Treatment

As “Motor Control”, “Segmental” and “Motor Control + Segmental” are all eligible “Exercise Training” programmes data needed to be extracted and presented for each Exercise Training Programme as the experimental intervention i.e

Motor Control vs Segmental

Motor Control vs Motor Control + Segmental

Motor Control vs No Treatment

Segmental vs Motor Control

Segmental vs Motor Control + Segmental 

Segmental vs No Treatment

Motor Control + Segmental vs Motor Control

Motor Control + Segmental vs Segmental

Motor Control + Segmental vs No Treatment

The reason we have duplicated some data but reversed the “Experimental” intervention is so the reader can see all trials relating to that exercise training programme visually within the forest plot. As the data is not pooled statistically duplication of data does not impact findings but does help the reader visualise the direction of effect for each exercise training programme. 

For example, if we only included Motor Control + Segmental vs Motor Control AND Motor Control + Segmental vs Segmental under motor control and segmental headings respectively, the reader would only see SMD data for Borisut Motor control vs No Treatment and Falla Motor control vs No treatment in the forest plot under the “Motor Control + Segmental” heading. This may lead them to believe “Motor Control + Segmental” exercise was only compared against no treatment and may miss that this exercise training programme also had large effects when compared against other types of exercise training. 

To aid in the clarify of this point we have done the following

1. Added a new table that includes basic interventions details for all trials. While a more detail version could originally be found in the Appendix, helping the reader identify the multiple treatment arms within trials will beneficial

2. We have added detail to the legends of the forest plot figures explaining how we have synthesised and presented the information

We also agree with the reviewer’s comments regarding structuring of results by comparison e.g motor control vs no treatment. An early draft of the manuscript had synthesised results in this way, however due to significant comparator heterogeneity there were multiple different groupings which meant synthesising similar trials was not possible and resulted in a description of individual trials. E.g for Motor Control ET Programmes the comparisons would have been

Motor Control vs No Treatment (1 trial)

Motor Control vs Usual Care (1 trial)

Motor Control vs AROM (1 trial)

Motor Control vs Segmental (2 trials)

Motor Control vs Pillar (4 trials)

Motor Control vs Motor Control & Segmental (1 trial)

Motor Control vs Proprioceptive Training (1 trial) 

As one of the main objectives of the systematic review was to synthesize evidence, we felt comparing exercise interventions to groups of comparators did not allow us to achieve this goal. 

To clarify this point to the reader we have added the following statement under “3.4 Results of individual studies and synthesis of results”

“Comparator heterogeneity also prevented the synthesize and analysis of evidence quality by contrasting each ET programme to subgroups of comparator interventions (e.g Motor Control vs No Treatment). Therefore, the effectiveness of each ET programme has been narratively described against all reported comparators.”

Furthermore, I notice you have only presented data in fig 4 (pain, short-term follow-up) and fig 5 (disability, short-term follow-up). It would be great, to see the summary synthesis for all outcomes and time-points data extracted, not only for short-term. A good suggestion here. We have now added figures for immediate term pain, intermediate pain and disability, long term pain.

Methods: Summary Measures – Please provide a reference and further clarification on the analysis approach used to calculate SMD and MD: change score or outcome analysis? In some studies it seem you have used outcome analysis (e.g. Borisut et al. 2013; Chung et al. 2018), and others you have used change scores (e.g. Bobos et al. 2016). Also, it would be great if you could mention the reason for the selection of approach and the software used to calculate SMD and MD with respective uncertainty interval. We would like to direct the reviewer to section 2.10 Synthesis of Results which states that post treatment means, and standard deviation were extracted owing to multiple trials not reporting change scores. This is the recommended approach by Cochrane which is referenced. The post treatment means and standard deviations were used for Bobos et al (2016) as this data was supplied directly by the author and identified as such in the table of characteristics appendix.

In the methods section we have outlined our plan to use Standardised mean difference where different measurement scales were used with 95% confidence intervals and mean difference where the same measurement scale was used as recommended by Cochrane (Section 2.9 Summary Measures). We have now added a reference to support this statement from the Cochrane handbook. We have also added into this statement that the calculations were completed using Review Manager 5.3

Due to considerable heterogeneity and to make it easier for the reader we have used SMD throughout and added the following under section 3.4 in the Results section

“Outcome measures heterogeneity limited summary measures to standardised mean differences plus 95% CI’s.”

Methods: Summary Measures – On line 195 you have stated that “Clinically importance differences were established a priori as a mean difference of 1/10 VAS and 5/50 NDI”. You have used Standard Mean Difference to estimate the effect for a few of the intervention contrast in this review. I was wondering if you could clarify that please? Furthermore, I was wondering what have you established as a clinically important difference for other outcomes such as Northwick Park Neck Pain Questionnaire and Neck and Shoulder Pain and Disability Index? We originally intended to use MD where trials used the same outcome measures and SMD were heterogeneity existed. 

Due to considerable heterogeneity and to make it easier for the reader we have used SMD throughout and added the following under section 3.4

“Outcome measures heterogeneity limited summary measures to standardised mean differences plus 95% CI’s.”

As per our protocol we had not established a clinically important difference for the Northwick Park Neck Pain Questionnaire or Neck and Shoulder Pain and Disability Index prior to completing our analysis. To our knowledge the clinically important difference of the Neck and Shoulder Pain and Disability Index is not reported within the literature. The Northwick Park Neck Pain Questionnaire clinically important difference is a 25%. We have reviewed the trials that used the Northwick Park Neck Pain Questionnaire and a clinically important difference was not found. As we have not planned to perform this analysis from our protocol and the results would not add any value we do not feel that establishing the clinical important difference for the Northwick Park Neck Pain Questionnaire is warranted.

Methods: Summary Measures – Following the previous comment, I was wondering if you could please clarify the use of Mean Difference (MD) for some studies and Standardised Mean Difference (SMD) for others. We originally intended to use MD where trials used the same outcome measures and SMD were heterogeneity existed. 

Due to considerable heterogeneity and to make it easier for the reader we have used SMD throughout and added the following under section 3.4

“Outcome measures heterogeneity limited summary measures to standardised mean differences plus 95% CI’s.”

Methods: Synthesis of results – To rate the quality of evidence you have used the Grading of Recommendations, Assessment, Development and Evaluation (GRADE) system. In S5 Appendix I can see you have pooled studies by outcome time-points (Immediate-term, Short-term, Intermediate-term, and Long-term). The overall certainty in the evidence should be assessed for each review outcome and for all studies pooled in the meta-analysis (pain-intensity and disability in this review). In this review it was not possible to pooled RCTs in a meta-analysis, so the GRADE should be assessed at individual RCT level for each intervention contrast and respective outcomes (e.g. “Motor Control vs No treatment” for both pain-intensity and disability). I suggest you review the quality of evidence assessment and modify S5 Appendix table accordingly. We agree with the reviewers comments that ideally GRADE should be assessed for each outcome by comparator. 

However due to comparator heterogeneity assessing the evidence in this way would not allow us to synthesize evidence which was the primary aim. 

To clarify this point to the reader we have added the following statement under “3.4 Results of individual studies and synthesis of results”

“Comparator heterogeneity also prevented the synthesis or analysis of evidence quality by contrasting each ET programme to subgroups of comparator interventions (e.g Motor Control vs No Treatment). Therefore, the effectiveness of each ET programme has been narratively described against all reported comparators.”

Results: Effectiveness of different exercise training programmes – I would suggest, if it is possible, to organise the narrative synthesis in results section by intervention contrast. It is just a suggestion as I believe it will help and probably be more clear for readers to see the effectiveness of the experimental intervention against comparator/control groups. I found it hard to follow.

For instance, you could have Motor Control vs No treatment:

- 3.5.1 Motor Control vs No treatment

The short-term effect of Motor control (MC) compared to No treatment on pain-intensity and disability was investigated in one trial (Borisul et al., 2013). There is xxx-quality evidence that ET involving MC is effective for pain-intensity (MD -1.17, 95% CI -1.78 to -0.57), and xxx-quality evidence that ET involving MC is effective for disability (MD -3.84, 95% CI -4.80 to -2.88) when compared to No treatment control. We believe and hope our edits from the points above have addressed this point to the reviewer’s satisfaction. 

Minor Comments 

Abstract: Objective – I suggest re-word objective to: “To synthesise the current evidence on the effectiveness of different ET programmes to reduce CNSNP and associated disability, and whether dosage affects outcomes.” Thank you for the suggestion. We have edited the manuscript accordingly.

Abstract: Results – I suggest you, whenever possible, to present the point estimate with respective uncertainty interval. Same for the result section. We understand the potential value of including point estimates however based on the heterogeneity, overall level of evidence we do not feel this would be appropriate or of value

Abstract: Results – Would be more clear if you could replace the term “Multiple” (in line 62) for the combination of ET programmes that reduced the outcome of pain-intensity and disability in the short-term. At the moment it is a bit confuse what multiple means. Thank you for this suggestion. We are unable to include each of the ET programmes within the abstract due to the word limitations. We have reworded this statement to provide more clarity

“A range of ET programmes reduce pain/disability in the short term (low to moderate evidence).”

Introduction: Please provide, a reference or further explanation/definition for what constituted an "Exercise Training (ET)" for this review. Thank you. We have updated the introduction to read

“The studies cited within these guidelines and systematic reviews describe multiple different exercise training (ET) programmes aimed at improving neuromuscular function or motor capacity of the neck and shoulder musculature.”

We have updated Section 2.1 Protocol and Registration

“The term resistance training has been changed from the protocol to “exercise training” as suggested by our patient and public involvement group to reflect exercise where the goal is to improve neuromuscular function or motor capacity of the neck and shoulder musculature.”

We have also updated Section 2.2.2 Intervention

“Interventions considered ET and included in this synthesis were exercises targeted at the neck or shoulders where an individual applies a force against resistance (gravity, their own hands, an external object) to improve neuromuscular function or motor capacity. Motor control exercises were included providing resistance was applied using a biofeedback unit or gravity”

Methods: Outcome Measures – I suggest to add the word “neck”. So the sentence reads: “Any patient reported measure of neck pain [e.g. Visual Analogue Scale (VAS)) and/or neck functional disability (e.g. Neck Disability Index, (NDI)].” Thank you. We have updated this accordingly. 

Methods: Synthesis of results – Maybe use the word narrative instead of quality in the following sentence on line 210. “Where meta-analysis was not possible, narrative synthesis provided summaries of the evidence.” Thank you. As per your previous comment we agree and have now updated the manuscript to read narrative synthesis where relevant

---

## [Editor Report · Decision Letter 1]

28 May 2020

Effectiveness and optimal dosage of exercise training for chronic non-specific neck pain: a systematic review with a narrative synthesis

PONE-D-20-07090R1

Dear Dr. Heneghan,

We are pleased to inform you that your manuscript has been judged scientifically suitable for publication and will be formally accepted for publication once it complies with all outstanding technical requirements.

With kind regards,

Jean-Philippe Regnaux, Ph.D, PT

Academic Editor

PLOS ONE

---

## [Editor Report · Acceptance letter]

1 Jun 2020

PONE-D-20-07090R1 

Effectiveness and optimal dosage of exercise training for chronic non-specific neck pain: a systematic review with a narrative synthesis 

Dear Dr. Heneghan:

I am pleased to inform you that your manuscript has been deemed suitable for publication in PLOS ONE. Congratulations! Your manuscript is now with our production department. 

With kind regards,

on behalf of

Mr Jean-Philippe Regnaux 

Academic Editor

PLOS ONE